# The Surprising Effectiveness of PPO in Cooperative Multi-Agent Games

**Chao Yu**[1♯*] **Akash Velu**[2♭*]**, Eugene Vinitsky**[2♭]**, Jiaxuan Gao**[1]**,**
**Yu Wang**[1♭]**, Alexandre Bayen**[2]**, Yi Wu**[13♭]
[1] Tsinghua University [2] University of California, Berkeley [3] Shanghai Qi Zhi Institute
♯zoeyuchao@gmail.com, ♭akashvelu@berkeley.edu

## Abstract

Proximal Policy Optimization (PPO) is a ubiquitous on-policy reinforcement learning algorithm but is significantly less utilized than off-policy learning algorithms in multi-agent settings. This is often due to the belief that PPO is significantly less sample efficient than off-policy methods in multi-agent systems. In this work, we carefully study the performance of PPO in cooperative multi-agent settings. We show that PPO-based multi-agent algorithms achieve surprisingly strong performance in four popular multi-agent testbeds: the particle-world environments, the StarCraft multi-agent challenge, Google Research Football, and the Hanabi challenge, with minimal hyperparameter tuning and without any domain-specific algorithmic modifications or architectures. Importantly, compared to competitive off-policy methods, PPO often achieves competitive or superior results in both final returns and sample efficiency. Finally, through ablation studies, we analyze implementation and hyperparameter factors that are critical to PPO's empirical performance, and give concrete practical suggestions regarding these factors. Our results show that when using these practices, simple PPO-based methods can be a strong baseline in cooperative multi-agent reinforcement learning. Source code is released at https://github.com/marlbenchmark/on-policy.

## 1 Introduction

Recent advances in reinforcement learning (RL) and multi-agent reinforcement learning (MARL) have led to a great deal of progress in creating artificial agents which can cooperate to solve tasks: DeepMind's AlphaStar surpassed professional-level performance in the StarCraft II [35], OpenAI Five defeated the world-champion in Dota II [4], and OpenAI demonstrated the emergence of human-like tool-use agent behaviors via multi-agent learning [2]. These notable successes were driven largely by on-policy RL algorithms such as IMPALA [10] and PPO [30, 4] which were often coupled with distributed training systems to utilize massive amounts of parallelism and compute. In the aforementioned works, tens of thousands of CPU cores and hundreds of GPUs were utilized to collect and train on an extraordinary volume of training samples. This is in contrast to recent academic progress and literature in MARL which has largely focused developing off-policy learning frameworks such as MADDPG [22] and value-decomposed Q-learning [32, 27]; methods in these frameworks have yielded state-of-the-art results on a wide range of multi-agent benchmarks [36, 37].

In this work, we revisit the use of *Proximal Policy Optimization* (PPO) – an on-policy algorithm[2] popular in single-agent RL but under-utilized in recent MARL literature – in multi-agent settings. We hypothesize that the relative lack of PPO in multi-agent settings can be attributed to two related factors: first, the belief that PPO is less sample-efficient than off-policy methods and is correspondingly less useful in resource-constrained settings, and second, the fact that common implementation and

---

*Equal Contribution. ♭ Equal Advising.

[2]Technically, PPO adopts off-policy corrections for sample-reuse. However, unlike off-policy methods, PPO does not utilize a replay buffer to train on samples collected throughout training.

hyperparameter tuning practices when using PPO in single-agent settings often do not yield strong performance when transferred to multi-agent settings.

We conduct a comprehensive empirical study to examine the performance of PPO on four popular cooperative multi-agent benchmarks: the multi-agent particle world environments (MPE) [22], the StarCraft multi-agent challenge (SMAC) [28], Google Research Football (GRF) [19] and the Hanabi challenge [3]. We first show that when compared to off-policy baselines, PPO achieves strong task performance and competitive sample-efficiency. We then identify five implementation factors and hyperparameters which are particularly important for PPO's performance, offer concrete suggestions about these configuring factors, and provide intuition as to why these suggestions hold.

Our aim in this work is *not* to propose a novel MARL algorithm, but instead to empirically demonstrate that with simple modifications, PPO can achieve strong performance in a wide variety of cooperative multi-agent settings. We additionally believe that our suggestions will assist practitioners in achieving competitive results with PPO.

Our contributions are summarized as follows:

- We demonstrate that PPO, without any domain-specific algorithmic changes or architectures and with minimal tuning, achieves final performances competitive to off-policy methods on four multi-agent cooperative benchmarks.
- We demonstrate that PPO obtains these strong results while using a comparable number of samples to many off-policy methods.
- We identify and analyze five implementation and hyperparameter factors that govern the practical performance of PPO in these settings, and offer concrete suggestions as to best practices regarding these factors.

## 2 Related Works

MARL algorithms generally fall between two frameworks: centralized and decentralized learning. Centralized methods [6] directly learn a single policy to produce the joint actions of all agents. In decentralized learning [21], each agent optimizes its reward independently; these methods can tackle general-sum games but may suffer from instability even in simple matrix games [12]. *Centralized training and decentralized execution (CTDE)* algorithms fall in between these two frameworks. Several past CTDE methods [22, 11] adopt actor-critic structures and learn a centralized critic which takes global information as input. Value-decomposition (VD) methods are another class of CTDE algorithms which represent the joint Q-function as a function of agents' local Q-functions [32, 27, 31] and have established state of the art results in popular MARL benchmarks [37, 36].

In single-agent continuous control tasks [8], advances in off-policy methods such as SAC [13] led to a consensus that despite their early success, policy gradient (PG) algorithms such as PPO are less sample efficient than off-policy methods. Similar conclusions have been drawn in multi-agent domains: [25] report that multi-agent PG methods such as COMA are outperformed by MADDPG and QMix [27] by a clear margin in the particle-world environment [23] and the StarCraft multi-agent challenge [28].

The use of PPO in multi-agent domains is studied by several concurrent works. [7] empirically show that decentralized, independent PPO (IPPO) can achieve high success rates in several hard SMAC maps – however, the reported IPPO results remain overall worse than QMix, and the study is limited to SMAC. [25] perform a broad benchmark of various MARL algorithms and note that PPO-based methods often perform competitively to other methods. Our work, on the other hand, focuses on PPO and analyzes its performance on a more comprehensive set of cooperative multi-agent benchmarks. We show PPO achieves strong results in the vast majority of tasks and also identify and analyze different implementation and hyperparameter factors of PPO which are influential to its performance multi-agent domains; to the best of our knowledge, these factors have not been studied to this extent in past work, particularly in multi-agent contexts.

Our empirical analysis of PPO's implementation and hyperparameter factors in multi-agent settings is similar to the studies of policy-gradient methods in single-agent RL [34, 17, 9, 1]. We find several of these suggestions to be useful and include them in our implementation. In our analysis, we focus on factors that are either largely understudied in the existing literature or are completely unique to the multi-agent setting.

# 3 PPO in Multi-Agent Settings

## 3.1 Preliminaries

We study decentralized partially observable Markov decision processes (DEC-POMDP) [24] with shared rewards. A DEC-POMDP is defined by $\langle \mathcal{S}, \mathcal{A}, O, R, P, n, \gamma \rangle$. $\mathcal{S}$ is the state space. $\mathcal{A}$ is the shared action space for each agent $i$. $o_i = O(s; i)$ is the local observation for agent $i$ at global state $s$. $P(s'|s, A)$ denotes the transition probability from $s$ to $s'$ given the joint action $A = (a_1, \ldots, a_n)$ for all $n$ agents. $R(s, A)$ denotes the shared reward function. $\gamma$ is the discount factor. Agents use a policy $\pi_\theta(a_i|o_i)$ parameterized by $\theta$ to produce an action $a_i$ from the local observation $o_i$, and jointly optimize the discounted accumulated reward $J(\theta) = \mathbb{E}_{A^t, s^t}\left[\sum_t \gamma^t R(s^t, A^t)\right]$ where $A^t = (a_1^t, \ldots, a_n^t)$ is the joint action at time step $t$.

## 3.2 MAPPO and IPPO

Our implementation of PPO in multi-agent settings closely resembles the structure of PPO in single-agent settings by learning a policy $\pi_\theta$ and a value function $V_\phi(s)$; these functions are represented as two separate neural networks. $V_\phi(s)$ is used for variance reduction and is only utilized during training; hence, it can take as input extra global information not present in the agent's local observation, allowing PPO in multi-agent domains to follow the CTDE structure. For clarity, we refer to PPO with centralized value function inputs as MAPPO (Multi-Agent PPO), and PPO with local inputs for both the policy and value function as IPPO (Independent PPO). We note that both MAPPO and IPPO operate in settings where agents share a common reward, as we focus only on cooperative settings.

## 3.3 Implementation Details

- **Parameter-Sharing**: In benchmark environments with homogeneous agents (i.e. agents have identical observation and action spaces), we utilize parameter sharing; past works have shown that this improves the efficiency of learning [5, 33], which is also consistent with our empirical findings. In these settings, agents share both the policy and value function parameters. A comparison of using parameter-sharing setting and learning separate parameters per agent can be found in Appendix C.2. We remark that agents are homogeneous in all benchmarks except for the *Comm* setting in the MPEs.
- **Common Implementation Practices:** We also adopt common practices in implementing PPO, including *Generalized Advantage Estimation (GAE)* [29] with advantage normalization and value-clipping. A full description of hyperparameter search settings, training details, and implementation details are in Appendix C. The source code for our implementation can be found in `https://github.com/marlbenchmark/on-policy`.

# 4 Main Results

## 4.1 Testbeds, Baselines, and Common Experimental Setup

**Testbed Environments:** We evaluate the performance of MAPPO and IPPO on four cooperative benchmark – the multi-agent particle-world environment (MPE), the StarCraft micromanagement challenge (SMAC), the Hanabi challenge, and Google Research Football (GRF) – and compare these methods' performance to popular off-policy algorithms which achieve state of the art results in each benchmark. Detailed descriptions of each testbed can be found in Appendix B.

**Baselines:** In each testbed, compare MAPPO and IPPO to a set of off-policy baselines, specifically:

- **MPEs**: QMix [27] and MADDPG [22].
- **SMAC**: QMix [27] and SOTA methods including QPlex [36], CWQMix [26], AIQMix [18] and RODE [37].
- **GRF**: QMix [27] and SOTA methods including CDS [20] and TiKick [16].
- **Hanabi**: SAD [15] and VDN [32].

**Common Experimental setup:** Here we give a brief description of the experimental setup common to all testbeds. Specific settings for each testbed are described later in Sec. 4.2-4.5.

- **Hyper-parameters Search**: For a fair comparison, we re-implement MADDPG and QMix and tune each method using a grid-search over a set of hyper-parameters such as learning rate,

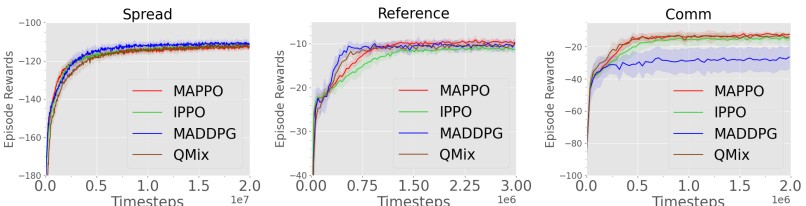

Figure 1: Performance of different algorithms in the MPEs.

target network update rate, and network architecture. We ensure that the size of this grid-search is equivalent to the size used to tune MAPPO and IPPO. We also test various relevant implementation tricks including value/reward normalization, hard and soft target network updates for Q-learning, and the input representation to the critic/mixer network.

- **Training Compute:** Experiments are performed on a desktop machine with 256 GB RAM, one 64-core CPU, and one GeForce RTX 3090 GPU used for forward action computation and training updates.

**Empirical Findings:** In the majority of environments, PPO achieves results better or comparable to the off-policy methods with ***comparable sample efficiency***.

## 4.2   MPE Testbed

**Experimental Setting:** We consider the three cooperative tasks proposed in [22]: the physical deception task (*Spread*), the simple reference task (*Reference*), and the cooperative communication task (*Comm*). As the MPE environment does not provide a global input, we follow [22] and concatenate all agents' local observations to form a global state which is utilized by MAPPO and the off-policy methods. Furthermore, *Comm* is the only task without homogenous agents; hence, we do not utilize parameter sharing for this task. All results are averaged over ten seeds.

**Experimental Results:** The performance of each algorithm at convergence is shown in Fig. 1. MAPPO achieves performance comparable and even superior to the off-policy baselines; we particularly see that MAPPO performs very similarly to QMix on all tasks and exceeds the performance of MADDPG in the *Comm* task, all while using a comparable number of environment steps. Despite not utilizing global information, IPPO also achieves similar or superior performance to centralized off-policy methods. Compared to MAPPO, IPPO converges to *slightly* lower final returns in several environments (*Comm* and *Reference*).

## 4.3   SMAC Testbed

**Experimental Setting:** We evaluate MAPPO with two different centralized value function inputs – labeled *AS* and *FP* – that combines agent-agnostic global information with agent-specific local information. These inputs are described fully in Section 5. All off-policy baselines utilize both the agent-agnostic global state and agent-specific local observations as input. Specifically, for agent $i$, the local Q-network (which computes actions at execution) takes in only the local agent-specific observation $o_i$ as input while the global mixer network takes in the agent-agnostic global state $s$ as input. For each random seed, we follow the evaluation metric proposed in [37]: we compute the win rate over 32 evaluation games after each training iteration and take the median of the final ten evaluation win-rates as the performance for each seed.

**Experimental Results:** We report the median win rates over six seeds in Table 1, which compares the PPO-based methods to QMix and RODE. Full results are deferred to Table 2 and Table 3 in Appendix. MAPPO, IPPO, and QMix are trained until convergence or reaching 10M environment steps. Results for RODE are obtained using the statistics from [37]. We observe that IPPO and MAPPO with both the *AS* and *FP* inputs achieve strong performance in the vast majority of SMAC maps. In particular, MAPPO and IPPO perform at least as well as QMix in most maps despite using the same number of samples. Comparing different value functions inputs, we observe that the performance of IPPO and MAPPO is highly similar, with the methods performing strongly in all but one map each. We also observe that MAPPO achieves performance comparable or superior to RODE's in 10 of 14 maps while using the same number of training samples. With more samples, the performance of MAPPO and IPPO continue to improve and ultimately match or exceed RODE's performance in nearly every

| Map | MAPPO(FP) | MAPPO(AS) | IPPO | QMix | RODE* | MAPPO*(FP) | MAPPO*(AS) |
|---|---|---|---|---|---|---|---|
| 2m vs_1z | **100.0**(0.0) | **100.0**(0.0) | **100.0**(0.0) | **95.3**(5.2) | / | _100.0_(0.0) | _100.0_(0.0) |
| 3m | **100.0**(0.0) | **100.0**(1.5) | **100.0**(0.0) | 96.9(1.3) | / | _100.0_(0.0) | _100.0_(1.5) |
| 2svs1sc | **100.0**(0.0) | **100.0**(0.0) | **100.0**(1.5) | 96.9(2.9) | _100.0_(0.0) | _100.0_(0.0) | _100.0_(0.0) |
| 2s3z | **100.0**(0.7) | **100.0**(1.5) | **100.0**(0.0) | 95.3(2.5) | _100.0_(0.0) | 96.9(1.5) | 96.9(1.5) |
| 3svs3z | **100.0**(0.0) | **100.0**(0.0) | **100.0**(0.0) | **96.9**(12.5) | / | _100.0_(0.0) | _100.0_(0.0) |
| 3svs4z | **100.0**(1.3) | **98.4**(1.6) | **99.2**(1.5) | **97.7**(1.7) | / | _100.0_(2.1) | _100.0_(1.5) |
| so many baneling | **100.0**(0.0) | **100.0**(0.7) | **100.0**(1.5) | 96.9(2.3) | / | _100.0_(1.5) | 96.9(1.5) |
| 8m | **100.0**(0.0) | **100.0**(0.0) | **100.0**(0.7) | 97.7(1.9) | / | _100.0_(0.0) | _100.0_(0.0) |
| MMM | **96.9**(0.6) | 93.8(1.5) | **96.9**(0.0) | 95.3(2.5) | / | _93.8_(2.6) | _96.9_(1.5) |
| 1c3s5z | **100.0**(0.0) | 96.9(2.6) | **100.0**(0.0) | 96.1(1.7) | _100.0_(0.0) | _100.0_(0.0) | 96.9(2.6) |
| bane vs bane | **100.0**(0.0) | **100.0**(0.0) | **100.0**(0.0) | **100.0**(0.0) | _100.0_(46.4) | _100.0_(0.0) | _100.0_(0.0) |
| 3svs5z | **100.0**(0.6) | **99.2**(1.4) | **100.0**(0.0) | 98.4(2.4) | 78.9(4.2) | _98.4_(5.5) | _100.0_(1.2) |
| 2cvs64zg | **100.0**(0.0) | **100.0**(0.0) | 98.4(1.3) | 92.2(4.0) | _100.0_(0.0) | _96.9_(3.1) | 95.3(3.5) |
| 8mvs9m | **96.9**(0.6) | **96.9**(0.6) | **96.9**(0.7) | 92.2(2.0) | / | _84.4_(5.1) | _87.5_(2.1) |
| 25m | **100.0**(1.5) | **100.0**(4.0) | **100.0**(0.0) | 85.9(7.1) | / | _96.9_(3.1) | _93.8_(2.9) |
| 5mvs6m | **89.1**(2.5) | **88.3**(1.2) | 87.5(2.3) | 75.8(3.7) | _71.1_(9.2) | 65.6(14.1) | 68.8(8.2) |
| 3s5z | **96.9**(0.7) | **96.9**(1.9) | **96.9**(1.5) | 88.3(2.9) | _93.8_(2.0) | 71.9(11.8) | 53.1(15.4) |
| 10mvs11m | **96.9**(4.8) | **96.9**(1.2) | 93.0(7.4) | **95.3**(1.0) | _95.3_(2.2) | 81.2(8.3) | _89.1_(5.5) |
| MMM2 | **90.6**(2.8) | 87.5(5.1) | 86.7(7.3) | 87.5(2.6) | _89.8_(6.7) | 51.6(21.9) | 28.1(29.6) |
| 3s5zvs3s6z | **84.4**(34.0) | 63.3(19.2) | **82.8**(19.1) | **82.8**(5.3) | _96.8_(25.11) | _75.0_(36.3) | 18.8(37.4) |
| 27mvs30m | **93.8**(2.4) | 85.9(3.8) | 69.5(11.8) | 39.1(9.8) | _96.8_(1.5) | _93.8_(3.8) | _89.1_(6.5) |
| 6hvs8z | **88.3**(3.7) | **85.9**(30.9) | 84.4(33.3) | 9.4(2.0) | _78.1_(37.0) | _78.1_(5.6) | _81.2_(31.8) |
| corridor | **100.0**(1.2) | 98.4(0.8) | 98.4(3.1) | 84.4(2.5) | _65.6_(32.1) | _93.8_(3.5) | _93.8_(2.8) |

Table 1: Median evaluation win rate and standard deviation on all the SMAC maps for different methods, Columns with "*" display results using the same number of timesteps as RODE. We bold all values within 1 standard deviation of the maximum and among the "*" columns, we denote all values within 1 standard deviation of the maximum with underlined italics. AS next to MAPPO indicates an agent-specific centralized input to the value function; FP indicates a similar agent-specific centralized input, but with redundant information removed.

map. As shown in Appendix D.1, MAPPO and IPPO perform comparably or superior to other other off-policy methods such as QPlex, CWQMix, and AIQMix in terms of both final performance and sample-efficiency.

Overall, MAPPO's effectiveness in nearly every SMAC map suggests that simple PPO-based algorithms can be strong baselines in challenging MARL problems.

## 4.4 Google Football Testbed

**Experimental Setting:** We evaluate MAPPO in several GRF academy scenarios, namely 3v.1, counterattack (CA) easy and hard, corner, pass-shoot (PS), and run-pass-shoot (RPS). In these scenarios, a team of agents attempts to score a goal against scripted opponent player(s). As the agents' local observations contain a full description of the environment state, there is no distinction between MAPPO and IPPO; for consistency, we label the results with PPO in Table 2 as "MAPPO". We utilize GRF's dense-reward setting in which all agents share a single reward which is the sum of individual agents' dense rewards. We compute the success rate over 100 rollouts of the game and report the average success rate over the last 10 evaluations, averaged over 6 seeds.

**Experimental Results:** We compare MAPPO with QMix and several SOTA methods, including CDS, a method that augments the environment reward with an intrinsic reward, and TiKick, an algorithm which combines online RL fine-tuning and large-scale offline pre-training. All methods except TiKick are trained for 25M environment steps in all scenarios with the exception of CA (hard) and Corner, in which methods are trained for 50M environment steps.

We generally observe in Table 2 that MAPPO achieves comparable or superior performance to other off-policy methods in *all* settings, despite not utilizing an intrinsic reward as is done in CDS. Comparing MAPPO to QMix, we observe that MAPPO clearly outperforms QMix in each scenario,

| Scen. | MAPPO | QMix | CDS | TiKick |
|---|---|---|---|---|
| 3v.1 | **88.03**(1.06) | 8.12(2.83) | 76.60(3.27) | 76.88(3.15) |
| CA(easy) | **87.76**(1.34) | 15.98(2.85) | 63.28(4.89) | / |
| CA(hard) | **77.38**(4.81) | 3.22(1.60) | 58.35(5.56) | 73.09(2.08) |
| Corner | **65.53**(2.19) | 16.10(3.00) | 3.80(0.54) | 33.00(3.01) |
| PS | **94.92**(0.68) | 8.05(3.66) | **94.15**(2.54) | / |
| RPS | **76.83**(1.81) | 8.08(4.71) | 62.38(4.56) | 79.12(2.06) |

Table 2: Average evaluation success rate and standard deviation (over six seeds) on GRF scenarios for different methods. All values within 1 standard deviation of the maximum success rate are marked in bold. We separate TiKick from the other methods as it uses pretrained models and thus does not constitute a direct comparison.

| # Players | Metric | MAPPO | IPPO | SAD | VDN |
|---|---|---|---|---|---|
| 2 | Avg. | 23.89(0.02) | **24.00**(0.02) | 23.87(0.03) | 23.83(0.03) |
| | Best | **24.23**(0.01) | 24.19(0.02) | 24.01(0.01) | 23.96(0.01) |
| 3 | Avg. | **23.77**(0.20) | 23.25(0.33) | 23.69(0.05) | 23.71(0.06) |
| | Best | **24.01**(0.01) | 23.87(0.03) | 23.93(0.01) | 23.99(0.01) |
| 4 | Avg. | **23.57**(0.13) | 22.52(0.37) | 23.27(0.26) | 23.03(0.15) |
| | Best | 23.71(0.01) | 23.06(0.03) | **23.81**(0.01) | 23.79(0.00) |
| 5 | Avg. | **23.04**(0.10) | 20.75(0.56) | 22.06(0.23) | 21.28(0.12) |
| | Best | **23.16**(0.01) | 22.54(0.02) | 23.01(0.01) | 21.80(0.01) |

Table 3: Best and Average evaluation scores of MAPPO, IPPO, SAD, and VDN on Hanabi-Full. Results are reported over at-least 3 seeds.

again while using the *same number of training samples*. MAPPO additionally outperforms TiKick on 4/5 scenarios, despite the fact that TiKick performs pre-training on a set of human expert data.

## 4.5 Hanabi Testbed

**Experimental Setting:** We evaluate MAPPO and IPPO in the full-scale Hanabi game with varying numbers of players (2-5 players). We compare MAPPO and IPPO to strong off-policy methods, namely Value Decomposition Networks (VDN) and Simplified Action Decoder (SAD), a Q-learning variant that has been successful in Hanabi. All methods do not utilize auxiliary tasks. Because each agent's local observation does not contain information about the agent's own cards[3], MAPPO utilizes a global-state that adds the agent's own cards to the local observation as input to its value function. VDN agents take only the local observations as input. SAD agents take as input not only the local observation provided by the environment, but also the greedy actions of other players in the past time steps (which is not used by MAPPO and IPPO). Due to algorithmic restrictions, no additional global information is utilized by SAD and VDN during centralized training. We follow [15] and report the average returns across at-least 3 random seeds as well as the best score achieved by any seed. The returns are averaged over 10k games.

**Experimental Results:** The reported results for SAD and VDN are obtained from [15]. All methods are trained for at-most 10B environment steps. As demonstrated in Table 3, MAPPO is able to produce results comparable or superior to the best and average returns achieved by SAD and VDN in nearly every setting, while utilizing the same number of environment steps. This demonstrates that even in environments such as Hanabi which require reasoning over other players' intents based on their actions, MAPPO can achieve strong performance, despite not explicitly modeling this intent.

IPPO's performance is comparable with MAPPO's in the 2-agent setting. However, as the agent number grows, MAPPO shows a clear margin of improvement over both IPPO and off-policy methods, which suggests that a centralized critic input can be crucial.

---

[3]The local observations in Hanabi contain information about the other agent's cards and game state.

# 5 Factors Influential to PPO's Performance

In this section, we analyze five factors that we find are especially influential to MAPPO's performance: value normalization, value function inputs, training data usage, policy/value clipping, and batch size. We find that these factors exhibit clear trends in terms of performance; using these trends, we give best-practice suggestions for each factor. We study each factor in a set of appropriate representative environments. All experiments are performed using MAPPO (i.e., PPO with centralized value functions) for consistency. Additional results can be found in Appendix E.

## 5.1 Value Normalization

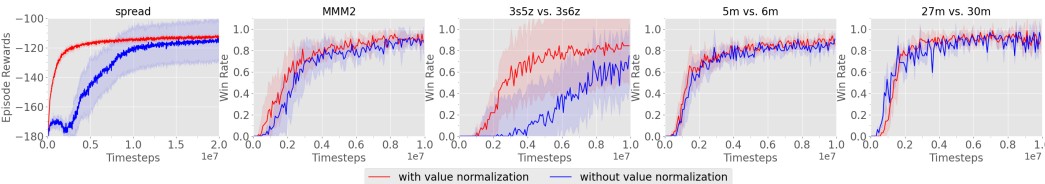

Figure 2: Impact of value normalization on MAPPO's performance in SMAC and MPE.

Through the training process of MAPPO, value targets can drastically change due to differences in the realized returns, leading to instability in value learning. To mitigate this issue, we standardize the targets of the value function by using running estimates of the average and standard deviation of the value targets. Concretely, during value learning, the value network regresses to normalized target values. When computing the GAE, we use the running average to denormalize the output of the value network so that the value outputs are properly scaled. We find that using value normalization never hurts training and often improves the final performance of MAPPO significantly.

**Empirical Analysis:** We study the impact of value-normalization in the MPE *spread* environment and several SMAC environments - results are shown in Fig. 2. In *Spread*, where the episode returns range from below -200 to 0, value normalization is critical to strong performance. Value normalization also has positive impacts on several SMAC maps, either by improving final performance or by reducing the training variance.

**Suggestion 1:** Utilize value normalization to stabilize value learning.

## 5.2 Input Representation to Value Function

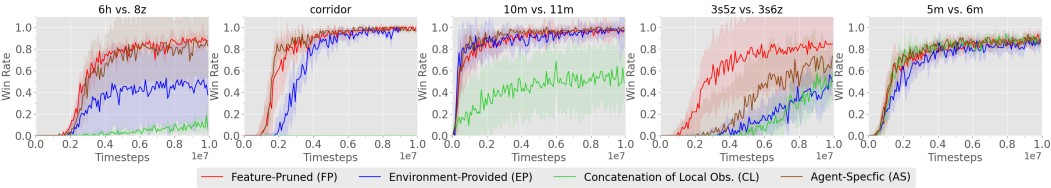

Figure 3: Effect of different value function input representations (described in Fig. 4).

The fundamental difference between many multi-agent CTDE PG algorithms and fully decentralized PG methods is the input to the value network. Therefore, the representation of the value input becomes an important aspect of the overall algorithm. The assumption behind using centralized value functions is that observing the full global state can make value learning easier. An accurate value function further improves policy learning through variance reduction.

Past works have typically used two forms of global states. [22] use a **concatenation of local observations (CL)** global state which is formed by concatenating all local agent observations. While it can be used in most environments, the *CL* state dimensionality grows with the number of agents and can omit important global information which is unobserved by all agents; these factors can make value learning difficult. Other works, particularly those studying SMAC, utilize an **Environment-Provided global state (EP)** which contains general global information about the environment state [11]. However, the *EP* state typically contains information common to all agents and can omit important local agent-specific information. This is true in SMAC, as shown in Fig. 4.

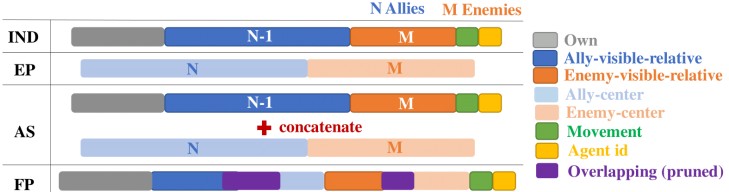

Figure 4: Different value function inputs with example features contained in each state (SMAC-specific). *IND* refers to using decentralized inputs (agents' local observations), *EP* refers to the environment provided global state, *AS* is an agent-specific global state which concatenates *EP* and *IND*, and *FP* is an agent-specific global state which prunes overlapping features from *AS*. *EP* omits important local data such as agent ID and available actions.

To address the weaknesses of the *CL* and *EP* states, we allow the value function to leverage both global and local information by forming an **Agent-Specific Global State (AS)** which creates a global state for agent $i$ by concatenating the *EP* state and $o_i$, the local observation for agent $i$. This provides the value function with a more comprehensive description of the environment state. However, if there is overlap in information between $o_i$ and the *EP* global state, then the *AS* state will have redundant information which unnecessarily increases the input dimensionality to the value function. As shown in Fig. 4, this is the case in SMAC. To examine the impact of this increased dimensionality, we create a **Featured-Pruned Agent-Specific Global State (FP)** by removing repeated features in the *AS* state.

**Emperical Analysis:** We study the impact of these different value function inputs in SMAC, which is the only considered benchmark that provides different options for centralized value function inputs. The results in Fig. 3 demonstrate that using the *CL* state, which is much higher dimensional than the other global states, is ineffective, particularly in maps with many agents. In comparison, using the *EP* global state achieves stronger performance but notably achieves subpar performance in more difficult maps, likely due to the lack of important local information. The *AS* and *FP* global states both achieve strong performance, with the *FP* state outperforming *AS* states on only several maps. This demonstrates that state dimensionality, agent-specific features, and global information are all important in forming an effective global state. We note that using the *FP* state requires knowledge of which features overlap between the *EP* state and the agents' local observations, and evaluate MAPPO with this state to demonstrate that limiting the value function input dimensionality can further improve performance.

**Suggestion 2:** When available, include both local, agent-specific features and global features in the value function input. Also check that these features do not unnecessarily increase the input dimension.

## 5.3   Training Data Usage

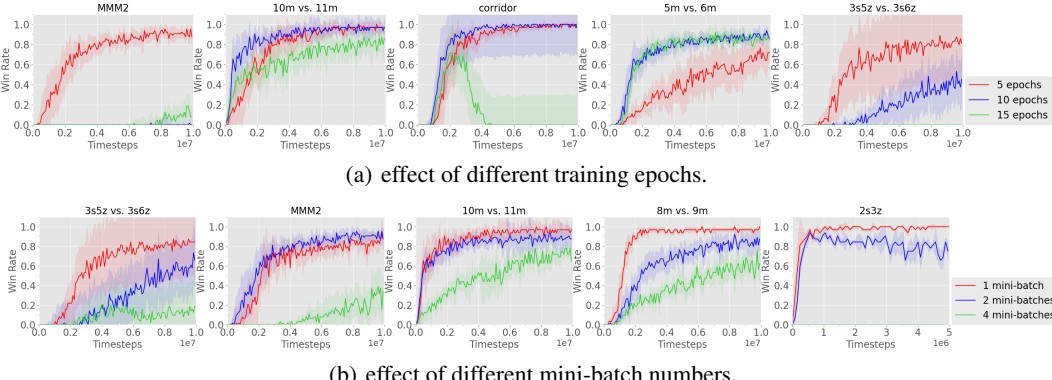

Figure 5: Effect of epoch and mini-batch number on MAPPO's performance in SMAC.

An important feature of PPO is the use of importance sampling for off-policy corrections, allowing sample reuse. [14] suggest splitting a large batch of collected samples into mini-batches and training for multiple epochs. In single-agent continuous control domains, the common practice is to split a large batch into about 32 or 64 mini-batches and train for tens of epochs. However, we find that in multi-agent domains, MAPPO's performance degrades when samples are re-used too often. Thus, we use 15 epochs for easy tasks, and 10 or 5 epochs for difficult tasks. We hypothesize that this

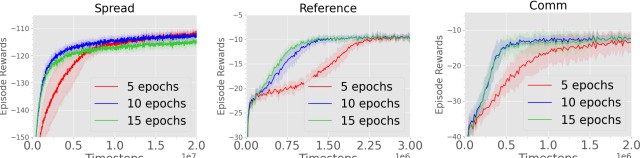

(a) effect of different training epochs.

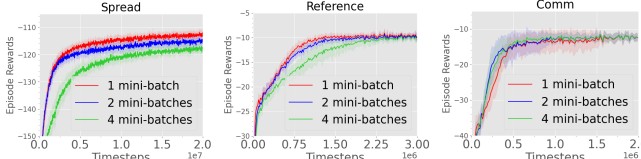

(b) effect of different mini-batch numbers.

Figure 6: Effect of epoch and mini-batch number on MAPPO's performance in MPE.

pattern could be a consequence of non-stationarity in MARL: using fewer epochs per update limits the change in the agents' policies, which could improve the stability of policy and value learning. Furthermore, similar to the suggestions by [17], we find that using more data to estimate gradients typically leads to improved practical performance. Thus, we split the training data into at-most two mini-batches and avoid mini-batching in the majority of situations.

**Experimental Analysis:** We study the effect of training epochs in SMAC maps in Fig. 5(a). We observe detrimental effects when training with large epoch numbers: when training with 15 epochs, MAPPO consistently learns a suboptimal policy, with particularly poor performance in the very difficult MMM2 and Corridor maps. In comparison, MAPPO performs well using 5 or 10 epochs. The performance of MAPPO is also highly sensitive to the number of mini-batches per training epoch. We consider three mini-batch values: 1, 2, and 4. A mini-batch of 4 indicates that we split the training data into 4 mini-batches to run gradient descent. Fig. 5(b) demonstrates that using more mini-batches negatively affects MAPPO's performance: when using 4 mini-batches, MAPPO fails to solve any of the selected maps while using 1 mini-batch produces the best performance on 22/23 maps. As shown in Fig. 6, similar conclusions can be drawn in the MPE tasks. In *Reference* and *Comm*, the simplest MPE tasks, all chosen epoch and minibatch values result in the same final performance, and using 15 training epochs even leads to faster convergence. However, in the harder *Spread* task, we observe a similar trend to SMAC: fewer epochs and no mini-batch splitting produces the best results.

**Suggestion 3:** Use at most 10 training epochs on difficult environments and 15 training epochs on easy environments. Additionally, avoid splitting data into mini-batches.

## 5.4 PPO Clipping

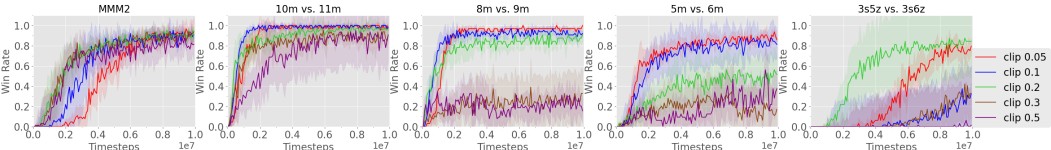

Figure 7: Effect of different clipping strengths on MAPPO's performance in SMAC.

Another core feature of PPO is the use of clipped importance ratio and value loss to prevent the policy and value functions from drastically changing between iterations. Clipping strength is controlled by the $\epsilon$ hyperparameter: large $\epsilon$ values allow for larger updates to the policy and value function. Similar to the number of training epochs, we hypothesize that policy and value clipping can limit the non-stationarity which is a result of the agents' policies changing during training. For small $\epsilon$, agents' policies are likely to change less per update, which we posit improves overall learning stability at the potential expense of learning speed. In single-agent settings, a common $\epsilon$ value is 0.2 [9, 1].

**Experimental Analysis:** We study the impact of PPO clipping strengths, controlled by the $\epsilon$ hyperparameter, in SMAC (Fig. 7). Note that $\epsilon$ is the same for both policy and value clipping. We generally that with small $\epsilon$ terms such as 0.05, MAPPO's learning speed is slowed in several maps, including hard maps such as MMM2 and 3s5z vs. 3s6z. However, final performance when using $\epsilon = 0.05$ is consistently high and the performance is more stable, as demonstrated by the smaller standard deviation in the training curves. We also observe that large $\epsilon$ terms such as 0.2, 0.3, and

0.5, which allow for larger updates to the policy and value function per gradient step, often result in sub-optimal performance.

**Suggestion 4:** For the best PPO performance, maintain a clipping ratio $\epsilon$ under 0.2; within this range, tune $\epsilon$ as a trade-off between training stability and fast convergence.

## 5.5 PPO Batch Size

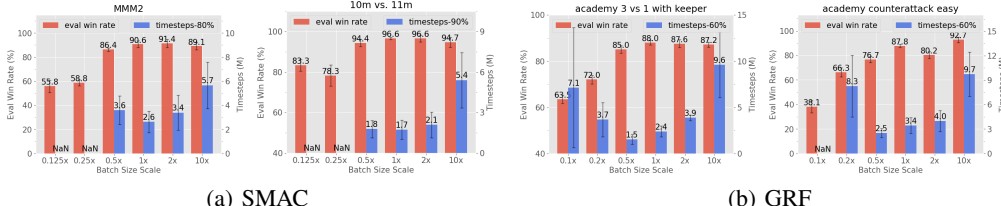

(a) SMAC                                              (b) GRF

Figure 8: Effect of batch size on MAPPO's performance in SMAC and GRF. Red bars show the final win-rates. The blue bars show the number of environment steps required to achieve a strong win-rate (80% or 90% in SMAC and 60% in GRF) as a measure of sample efficiency. "NaN" means such a win-rate was never reached. The x-axis specifies the batch-size as a multiple of the batch-size used in our main results. A sufficiently large batch-size is required to achieve the best final performance/sample efficiency; further increasing the batch size may hurt sample efficiency.

During training updates, PPO samples a batch of on-policy trajectories which are used to estimate the gradients for the policy and value function objectives. Since the number of mini-batches is fixed in our training (see Sec. 5.3), a larger batch generally will result in more accurate gradients, yielding better updates to the value functions and policies. However, the accumulation of the batch is constrained by the amount of available compute and memory: collecting a large set of trajectories requires extensive parallelism for efficiency and the batches need to be stored in GPU memory. Using an unnecessarily large batch-size can hence be wasteful in terms of required compute and sample-efficiency.

**Experimental Analysis:** The impact of various batch sizes on both final task performance and sample-efficiency is demonstrated in Fig. 8. We observe that in nearly all cases, there is a critical batch-size setting - when the batch-size is below this critical point, the final performance of MAPPO is poor, and further tuning the batch size produces the optimal final performance and sample-efficiency. However, continuing to increase the batch size may not result in improved final performance and in-fact can worsen sample-efficiency.

**Suggestion 5:** Utilize a large batch size to achieve best task performance with MAPPO. Then, tune the batch size to optimize for sample-efficiency.

## 6 Conclusion

This work demonstrates that PPO, an on-policy policy gradient RL algorithm, achieves strong results in both final returns and sample efficiency that are comparable to the state-of-the-art methods on a variety of cooperative multi-agent challenges, which suggests that properly configured PPO can be a competitive baseline for cooperative MARL tasks. We also identify and analyze five key implementation and hyperparameter factors that are influential in PPO's performance in these settings. Based on our empirical studies, we give concrete suggestions for the best practices with respect to these factors. There are a few limitations in this work that point to directions for future study. Firstly, our benchmark environments all use discrete action spaces, are all cooperative, and in the vast majority of cases, contain homogeneous agents. In future work, we aim to test PPO on a wider range of domains such as competitive games and MARL problems with continuous action spaces and heterogeneous agents. Furthermore, our work is primarily empirical in nature, and does not directly analyze the theoretical underpinnings of PPO. We believe that the empirical analysis of our suggestions can serve as starting points for further analysis into PPO's properties in MARL.

### Acknowledgments

This research is supported by NSFC (U20A20334, U19B2019 and M-0248), Tsinghua-Meituan Joint Institute for Digital Life, Tsinghua EE Independent Research Project, Beijing National Research Center for Information Science and Technology (BNRist), Beijing Innovation Center for Future Chips and 2030 Innovation Megaprojects of China (Programme on New Generation Artificial Intelligence) Grant No. 2021AAA0150000.

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
