# Supplementary Materials of
# The Surprising Effectiveness of PPO in Cooperative Multi-Agent Games

**Chao Yu[1][♯][*], Akash Velu[2][♮][*], Eugene Vinitsky[2][♭], Jiaxuan Gao[1],**
**Yu Wang[1][♭], Alexandre Bayen[2], Yi Wu[13][♭]**
[1] Tsinghua University [2] University of California, Berkeley [3] Shanghai Qi Zhi Institute
[♯]zoeyuchao@gmail.com, [♮]akashvelu@berkeley.edu

## A    MAPPO Details

MAPPO trains two separate neural networks: an actor network with parameters $\theta$, and a value function network (referred to as a critic) with parameters $\phi$. These networks can be shared amongst all agents if the agents are homogeneous, but each agent can also have its own pair of actor and critic networks. We assume here that all agents share critic and actor networks, for notational convenience. Specifically, the critic network, denoted as $V_\phi$, performs the following mapping: $S \to \mathbb{R}$. The global state can be agent-specific or agent-agnostic.

The actor network, denoted as $\pi_\theta$, maps agent observations $o_t^{(a)}$ to a categorical distribution over actions in discrete action spaces, or to the mean and standard deviation vectors of a Multivariate Gaussian Distribution, from which an action is sampled, in continuous action spaces.

The actor network is trained to maximize the objective

$L(\theta) = [\frac{1}{Bn} \sum_{i=1}^{B} \sum_{k=1}^{n} \min(r_{\theta,i}^{(k)} A_i^{(k)}, \text{clip}(r_{\theta,i}^{(k)}, 1-\epsilon, 1+\epsilon) A_i^{(k)})] + \sigma \frac{1}{Bn} \sum_{i=1}^{B} \sum_{k=1}^{n} S[\pi_\theta(o_i^{(k)}))]$, where

$r_{\theta,i}^{(k)} = \frac{\pi_\theta(a_i^{(k)}|o_i^{(k)})}{\pi_{\theta_{old}}(a_i^{(k)}|o_i^{(k)})}$. $A_i^{(k)}$ is computed using the GAE method, $S$ is the policy entropy, and $\sigma$ is the entropy coefficient hyperparameter.

The critic network is trained to minimize the loss function

$L(\phi) = \frac{1}{Bn} \sum_{i=1}^{B} \sum_{k=1}^{n} (\max[(V_\phi(s_i^{(k)}) - \hat{R}_i)^2, (\text{clip}(V_\phi(s_i^{(k)}), V_{\phi_{old}}(s_i^{(k)}) - \varepsilon, V_{\phi_{old}}(s_i^{(k)}) + \varepsilon) - \hat{R}_i)^2],$

where $\hat{R}_i$ is the discounted reward-to-go.

In the loss functions above, $B$ refers to the batch size and $n$ refers to the number of agents.

If the critic and actor networks are RNNs, then the loss functions additionally sum over time, and the networks are trained via Backpropagation Through Time (BPTT). Pseudocode for recurrent-MAPPO is shown in Alg. 1.

## B    Testing domains

**Multi-agent Particle-World Environment (MPE)** was introduced in (Lowe et al., 2017). MPE consist of various multi-agent games in a 2D world with small particles navigating within a square box. We consider the 3 fully cooperative tasks from the original set shown in Fig. 1(a): *Spread*, *Comm*, and *Reference*. Note that since the two agents in *speaker-listener* have different observation

---

[*]Equal Contribution. [♭] Equal Advising.

**Algorithm 1** Recurrent-MAPPO

---

Initialize $\theta$, the parameters for policy $\pi$ and $\phi$, the parameters for critic $V$, using Orthogonal initialization (Hu et al., 2020)
Set learning rate $\alpha$
**while** $step \leq step_{\max}$ **do**
    set data buffer $D = \{\}$
    **for** $i = 1$ **to** $batch\_size$ **do**
        $\tau = []$ empty list
        initialize $h_{0,\pi}^{(1)}, \ldots h_{0,\pi}^{(n)}$ actor RNN states
        initialize $h_{0,V}^{(1)}, \ldots h_{0,V}^{(n)}$ critic RNN states
        **for** $t = 1$ **to** $T$ **do**
            **for all** agents $a$ **do**
                $p_t^{(a)}, h_{t,\pi}^{(a)} = \pi(o_t^{(a)}, h_{t-1,\pi}^{(a)}; \theta)$
                $u_t^{(a)} \sim p_t^{(a)}$
                $v_t^{(a)}, h_{t,V}^{(a)} = V(s_t^{(a)}, h_{t-1,V}^{(a)}; \phi)$
            **end for**
            Execute actions $\boldsymbol{u_t}$, observe $r_t, s_{t+1}, \boldsymbol{o_{t+1}}$
            $\tau += [s_t, \boldsymbol{o_t}, \boldsymbol{h_{t,\pi}}, \boldsymbol{h_{t,V}}, \boldsymbol{u_t}, r_t, s_{t+1}, \boldsymbol{o_{t+1}}]$
        **end for**
        Compute advantage estimate $\hat{A}$ via GAE on $\tau$, using PopArt
        Compute reward-to-go $\hat{R}$ on $\tau$ and normalize with PopArt
        Split trajectory $\tau$ into chunks of length L
        **for** $l = 0, 1, .., $ T//L **do**
            $D = D \cup (\tau[l : l + T], \hat{A}[l : l + L], \hat{R}[l : l + L])$
        **end for**
    **end for**
    **for** mini-batch $k = 1, \ldots, K$ **do**
        $b \leftarrow$ random mini-batch from D with all agent data
        **for** each data chunk $c$ in the mini-batch $b$ **do**
            update RNN hidden states for $\pi$ and $V$ from first hidden state in data chunk
        **end for**
    **end for**
    Adam update $\theta$ on $L(\theta)$ with data $b$
    Adam update $\phi$ on $L(\phi)$ with data $b$
**end while**

---

and action spaces, this is the only setting in this paper where we do not share parameters but train separate policies for each agent.

**StarCraftII Micromanagement Challenge (SMAC)** tasks were introduced in (Rashid et al., 2019). In these tasks, decentralized agents must cooperate to defeat adversarial bots in various scenarios with a wide range of agent numbers (from 2 to 27). We use a global game state to train our centralized critics or Q-functions. Fig. 1(c) and 1(d) show two example StarCraftII environments.

As described in Sec. 5.2, we utilize an agent-specific global state as input to the global state. This agent-specific global state augments the original global state provided by the SMAC environment by adding relevant agent-specific features.

Specifically, the original global state of SMAC contains information about all agents and enemies - this includes information such as the distance from each agent/enemy to the map center, the health of each agent/enemy, the shield status of each agent/enemy, and the weapon cooldown state of each agent. However, when compared to the local observation of each agent, the global state does not contain agent-specific information including agent id, agent movement options, agent attack options, relative distance to allies/enemies. Note that the local observation contains information only about allies/enemies within a sight radius of the agent. To address the lack of critical local information in the environment provided global state, we create several other global inputs which are specific to each agent, and combine local and global features. The first, which we call *agent-specific (AS)*, uses

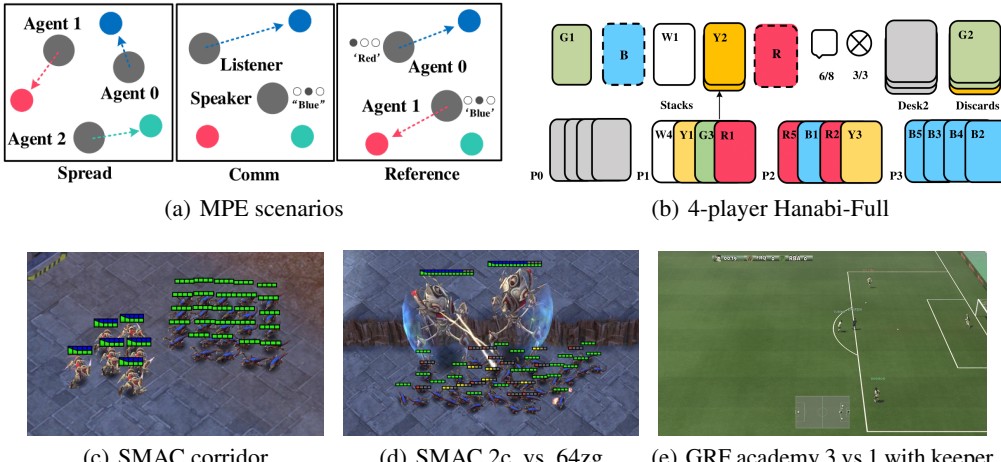

(a) MPE scenarios        (b) 4-player Hanabi-Full

(c) SMAC corridor     (d) SMAC 2c_vs_64zg     (e) GRF academy 3 vs 1 with keeper

Figure 1: Task visualizations. (a) The MPE domain. *Spread* (left): agents need to cover all the landmarks and do not have a color preference for the landmark they navigate to; *Comm* (middle): the listener needs to navigate to a specific landmarks following the instruction from the speaker; *Reference* (right): both agents only know the other's goal landmark and needs to communicate to ensure both agents move to the desired target. (b) The Hanabi domain: 4-player *Hanabi-Full* - figure obtained from (Bard et al., 2020). (c) The *corridor* map in the SMAC domain. (d) The *2c vs. 64zg* map in the SMAC domain. (e) The *academy 3 vs 1 with keeper* scenario in the GRF domain.

the concatenation of the environment provided global state and agent *i*'s observation, $o_i$, as the global input to MAPPO's critic during gradient updates for agent *i*. However, since the global state and local agent observations have overlapping features, we additionally create a feature-pruned global state (*FP*) which removes the overlapping features in the *AS* global state.

**Hanabi** is a turn-based card game, introduced as a MARL challenge in (Bard et al., 2020) , where each agent observes other players' cards except their own cards. A visualization of the game is shown in Fig. 1(b). The goal of the game is to send information tokens to others and cooperatively take actions to stack as many cards as possible in ascending order to collect points.

The turn-based nature of Hanabi presents a challenge when computing the reward for an agent during it's turn. We utilize the forward accumulated reward as one turn reward $R_i$; specifically, if there are 4 players and players 0, 1, 2, and 3 execute their respective actions at timesteps *k, k+1, k+2, k+3* respectively, resulting in rewards of $r_k^{(0)}, r_{k+1}^{(1)}, r_{k+2}^{(2)}, r_{k+3}^{(3)}$, then the reward assigned to player 0 will be $R_0 = r_k^{(0)} + r_{k+1}^{(1)} + r_{k+2}^{(2)} + r_{k+3}^{(3)}$ and similarly, the reward assigned to player 1 will be $R_1 = r_{k+1}^{(1)} + r_{k+2}^{(2)} + r_{k+3}^{(3)} + r_{k+4}^{(0)}$. Here, $r_t^i$ denotes the reward received at timestep *t* when agent *i* is executes a move.

**Google Research Football (GRF)**, introduced in [3], contains a set of cooperative multi-agent challenges in which a team of agents play a team of bots in various football scenarios. In the scenarios we consider, the goal of the agents is to score a goal against the opposing team. Fig. 1(e) shows the example academy scenario.

The agents' local observations contain a complete description of the environment state at any given time; hence, both the policy and value-function take as input the same observation. At each step, agents share the same reward $R_t$, which is computed as the sum of per-agent rewards $r_t^{(i)}$ which represents the progress made by agent *i*.

## C    Training details

### C.1    Implementation

All algorithms utilize parameter sharing - i.e., all agents share the same networks - in all environments except for the *Comm* scenario in the MPE. Furthermore, we tune the architecture and hyperparameters

| Map | MAPPO | MAPPO-Ind |
|---|---|---|
| 1c3s5z | **100.0(0.0)** | 99.1(0.7) |
| 2s3z | **100.0(0.7)** | 99.1(0.9) |
| 3s_vs_5z | **100.0(0.6)** | 93.8(1.8) |
| 3s5z | **96.9(0.7)** | 80.4(3.3) |
| 3s5z_vs_3s6z | 84.4(34.0) | 37.8(5.6) |
| 5m_vs_6m | **89.1(2.5)** | 44.4(2.9) |
| 6h_vs_8z | **88.3(3.7)** | 11.4(2.5) |
| 10m_vs_11m | **96.9(4.8)** | 78.4(2.7) |
| corridor | **100.0(1.2)** | 82.2(1.8) |
| MMM2 | 90.6(2.8) | 13.0(3.7) |

Table 1: Median evaluation win rate (standard deviation) on selected SMAC maps over 6 random seeds.

of MADDPG and QMix, and thus use different hyperparameters than the original implementations. However, we ensure that the performance of the algorithms in the baselines matches or exceeds the results reported in their original papers.

For each algorithm, certain hyperparameters are kept constant across all environments; these are listed in Tables 4 and 5 for MAPPO, QMix, and MADDPG, respectively. These values are obtained either from the PPO baselines implementation in the case of MAPPO, or from the original implementations for QMix and MADDPG. Note that since we use parameter sharing and combine all agents' data, the actual batch-sizes will be larger with more agents.

In these tables, "recurrent data chunk length" refers to the length of chunks that a trajectory is split into before being used for training via BPTT (only applicable for RNN policies). "Max clipped value loss" refers to the value-clipping term in the value loss. "Gamma" refers to the discount factor, and "huber delta" specifies the delta parameter in the Huber loss function. "Epsilon" describes the starting and ending value of $\epsilon$ for $\epsilon$-greedy exploration, and "epsilon anneal time" refers to the number of environment steps over which $\epsilon$ will be annealed from the starting to the ending value, in a linear manner. "Use feature normalization" refers to whether the feature normalization is applied to the network input.

## C.2 Parameter Sharing

In the main results which are presented, we utilize parameter sharing - a technique which has been shown to be beneficial in a variety of state-of-the-art methods [1, 4] in all algorithms for a fair comparison. Specifically, both the policy and value network parameters are shared across all agents. In this appendix section, we include results which demonstrate the benefit of parameter sharing. Table 1 shows median evaluation win rate (with standard deviation in paranttheses) on selected SMAC maps over 6 random seeds. MAPPO-Ind is MAPPO denotes MAPPO without parameter sharing - e.g., each agent has a separate policy and value function network. We observe that MAPPO with parameter sharing outperforms MAPPO without parameter sharing by a clear margin, supporting our decision to adopt parameter sharing in all PPO experiments and all baselines used in our results. A more theoretical analysis of the effect of parameter sharing can be found in [2].

## C.3 Death Masking

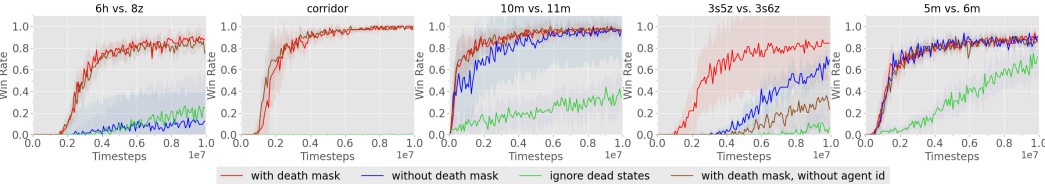

Figure 2: The effect of death mask on MAPPO's performance in SMAC.

In SMAC, it is possible through the course of an episode for certain agents to become inactive, or "die" while other agents remain active in the environment. In this setting, while the local observation for a dead agent becomes all zeros except for the agent's ID, the value-state still contains other nonzero features about the environment. When computing the GAE for an agent during training, it is unclear how to handle timesteps in which the agent is dead. We consider four options: (1) in which we replace the value state for a dead agent with a zero state containing the agent ID (similar to it's local observation). We refer to this as "death masking"; (2) MAPPO without death masking, i.e., still using the nonzero global state as value input; (3) completely drop the transition samples after an agent dies (note that we still need to accumulate rewards after the agent dies to correctly estimate episode returns); and (4) replacing the global state with a pure zero-state which does not include the agent ID. Fig. 2 demonstrates that variant (1) significantly outperforms variants (2) and (3), and consistently achieves overall strong performance. Including the agent id in the death mask, as is done in variant (1), is particularly important in maps which agents may take on different roles, as demonstrated by the superior performance of variant (1) compared to variant (4), which does not contain the agent ID in the death-mask zero-state, in the 3s5z vs. 3s6z map.

**Justification of Death Masking** Let $\mathbf{0}_a$ be a zero vector with agent $a$'s agent ID appended to the end. The use of agent ID leads to an agent-specific value function depending on an agent's type or role. It has been empirically justified that such an agent-specific feature is particularly helpful when the environment contains heterogeneous agents.

We now provide some intuition as to why using $\mathbf{0}_a$ as the critic input when agents are dead appears to be a better alternative to using the usual agent-specific global state as the input to the value function. Note that our global state to the value network has agent-specific information, such as available actions and relative distances to other agents. When an agent dies, these agent-specific features become zero, while the remaining agent-agnostic features remain nonzero - this leads to a drastic distribution shift in the critic input compared to states in which the agent is alive. In most SMAC maps, an agent is dead in only a small fraction of the timesteps in a batch (about 20%); due to their relative infrequency in the training data the states in which an agent is dead will likely have large value prediction error. Moreover, it is also possible that training on these out of distribution inputs harms the feature representation of the value network.

Although replacing the states at which an agent is dead with a fixed vector $\mathbf{0}_a$ also results in a distribution shift, the replacement results in there being only 1 vector which captures the state at which an agent is dead - thus, the critic is more likely to be able to fit the average post-death reward for agent $a$ to the input $\mathbf{0}_a$. Our ablation on the value function fitting error provide some weight to this hypothesis.

Another possible mechanism of handling agent deaths is to completely skip value learning in states in which an agent is dead, by essentially terminating an agent's episode when it dies. Suppose the game episode is $T$ and the agent dies at timestep $d$. If we are not learning on dead state then, in order to correctly accumulate the episode return, we need to replace the reward $r_d$ at timestep $d$ by the total return $R_d$ at time $d$, i.e., $r_d \leftarrow R_d = \sum_{t=d}^{T} \gamma^{t-d} r_t$. We would then need to compute the GAE only on those states in which the agent is alive. While this approach is theoretically correct (we are simply treating the state where the agent died as a terminal state and assigning the accumulated discounted reward as a terminal reward), it can have negative ramifications in the policy learning process, as outlined below.

The GAE is an exponentially weighted average of $k$-step returns intended to trade off between bias and variance. Large $k$ values result in a low bias, but high variance return estimate, whereas small $k$ values result in a high bias, low variance return estimate. However, since the entire post death return $R_d$ replaces the single timestep reward $r_d$ at timestep $d$, computing the 1-step return estimate at timestep $d$ essentially becomes a $(T-d)$-step estimate, eliminating potential benefits of value function truncation of the trajectory and potentially leading to higher variance. This potentially dampens the benefit that could come from using the GAE at the timesteps in which an agent is dead.

We analyze the impact of the death masking by comparing different ways of handling dead agents, including: (1) our death masking, (2) using global states without death masking and (3) ignoring dead states in value learning and in the GAE computation. We first examine the median win rate with these different options in Fig. 11 and 13. It is evident that our method of death masking, which uses $\mathbf{0}_a$ as the input to the critic when an agent is dead, results in superior performance compared to other options.

Fig. 14 also demonstrates that using the death mask results in a lower values loss in the vast majority of SMAC maps, demonstrating that the accuracy of the value predictions improve when using the death mask. While the arguments here are intuitive the clear experimental benefits suggest that theoretically characterizing the effect of this method would be valuable.

### C.4 Hyperparameters

Tables 4-16 describe the common hyperparameters, hyperparameter grid search values, and chosen hyperparmeters for MAPPO, QMix, and MADDPG in all testing domains. Tables 6, 7, 8, and 9 describe common hyperparameters for different algorithms in each domain. Tables 10, 11, and 12 describe the hyperparameter grid search procedure for the MAPPO, QMix, and MADDPG algorithms, respectively. Lastly, Tables 13, 14, 15 and 16 describe the final chosen hyperparameters among fine-tuned parameters for different algorithms in MPE, SMAC, Hanabi, and GRF, respectively.

For MAPPO, "Batch Size" refers to the number of environment steps collected before updating the policy via gradient descent. Since agents do not share a policy only in the MPE speaker-listener, the batch size does not depend on the number of agents in the speaker-listener environment. "Mini-batch" refers to the number of mini-batches a batch of data is split into, "gain" refers to the weight initialization gain of the last network layer for the actor network. "Entropy coef" is the entropy coefficient $\sigma$ in the policy loss. "Tau" corresponds to the rate of the polyak average technique used to update the target networks, and if the target networks are not updated in a "soft" manner, the "hard interval" hyperparameter specifies the number of gradient updates which must elapse before the target network parameters are updated to equal the live network parameters. "Clip" refers to the $\epsilon$ hyperparameter in the policy objective and value loss which controls the extent to which large policy and value function changes are penalized.

MLP network architectures are as follows: all MLP networks use "num fc" linear layers, whose dimensions are specified by the "fc layer dim" hyperparameter. When using MLP networks, "stacked frames" refers to the number of previous observations which are concatenated to form the network input: for instance, if "stacked frames" equals 1, then only the current observation is used as input, and if "stacked frames" is 2, then the current and previous observations are concatenated to form the input. For RNN networks, the network architecture is "num fc" fully connected linear layers of dimension "fc layer dim", followed by "num GRU layers" GRU layers, finally followed by "num fc after" linear layers.

## D  Additional Results

### D.1  Additional SMAC Results

Results of all algorithms in all SMAC maps can be found in Tab. 2 and 3.

As MAPPO does not converge within 10M environment steps in the 3s5z vs. 3s6z map, Fig. 3 shows the performance of MAPPO in 3s5z vs. 3s6z when run until convergence. Fig. 4 presents the evaluation win of MAPPO with different value inputs (*FP* and *AS*), decentralized PPO (IPPO), QMix, and QMix with a modified global state input to the mixer network, which we call QMix (MG). Specifically, QMix(MG) uses a concatenation of the default environment global state, as well as *all* agents' local observations, as the mixer network input.

Fig. 5 compares the results of MAPPO(FP) to various off-policy baselines, including QMix(MG), RODE, QPLEX, CWQMix, and AIQMix, in many SMAC maps. Both QMIX and RODE utilize both the agent-agnostic global state and agent-specific local observations as input. Specifically, for agent $i$, the local Q-network (which computes actions at execution) takes in only the local agent-specific observation $o_i$ as input while the global mixer network takes in the agent-agnostic global state $s$ as input. This is also the case for the other value-decomposition methods presented in Appendix Table 1 (QPLEX, CWQMix, and AIQMix).

### D.2  Additional GRF Results

Fig. 6 compares the results of MAPPO to various baselines, including QMix, CDS, and TiKick, in 6 academy scenarios.

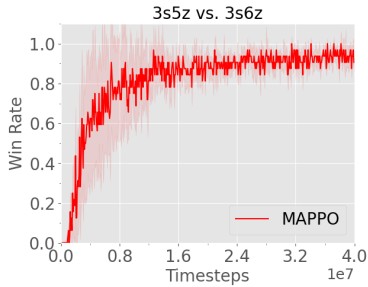

Figure 3: Median win rate of 3s5z vs. 3s6z map after 40M environment steps.

# E  Ablation Studies

We present the learning curves for all ablation studies performed. Fig. 7 demonstrates the impact of value normalization on MAPPO's performance. Fig. 8 shows the effect of global state information on MAPPO's performance in SMAC. Fig. 9 studies the influence of training epochs on MAPPO's performance. Fig. 10 studies the influence of clipping term on MAPPO's performance. Fig. 11 and Fig. 12 illustrates the influence of the death mask on MAPPO(FP)'s and MAPPO(AS)'s performance. Similarly, Fig. 13 compares the performance of MAPPO when ignoring states in which an agent is dead when computing GAE to using the death mask when computing the GAE. Fig. 14 illustrates the effect of death mask on MAPPO's value loss in the SMAC domain. Lastly, Fig. 15 shows the influence of including the agent-id in the agent-specific global state.

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

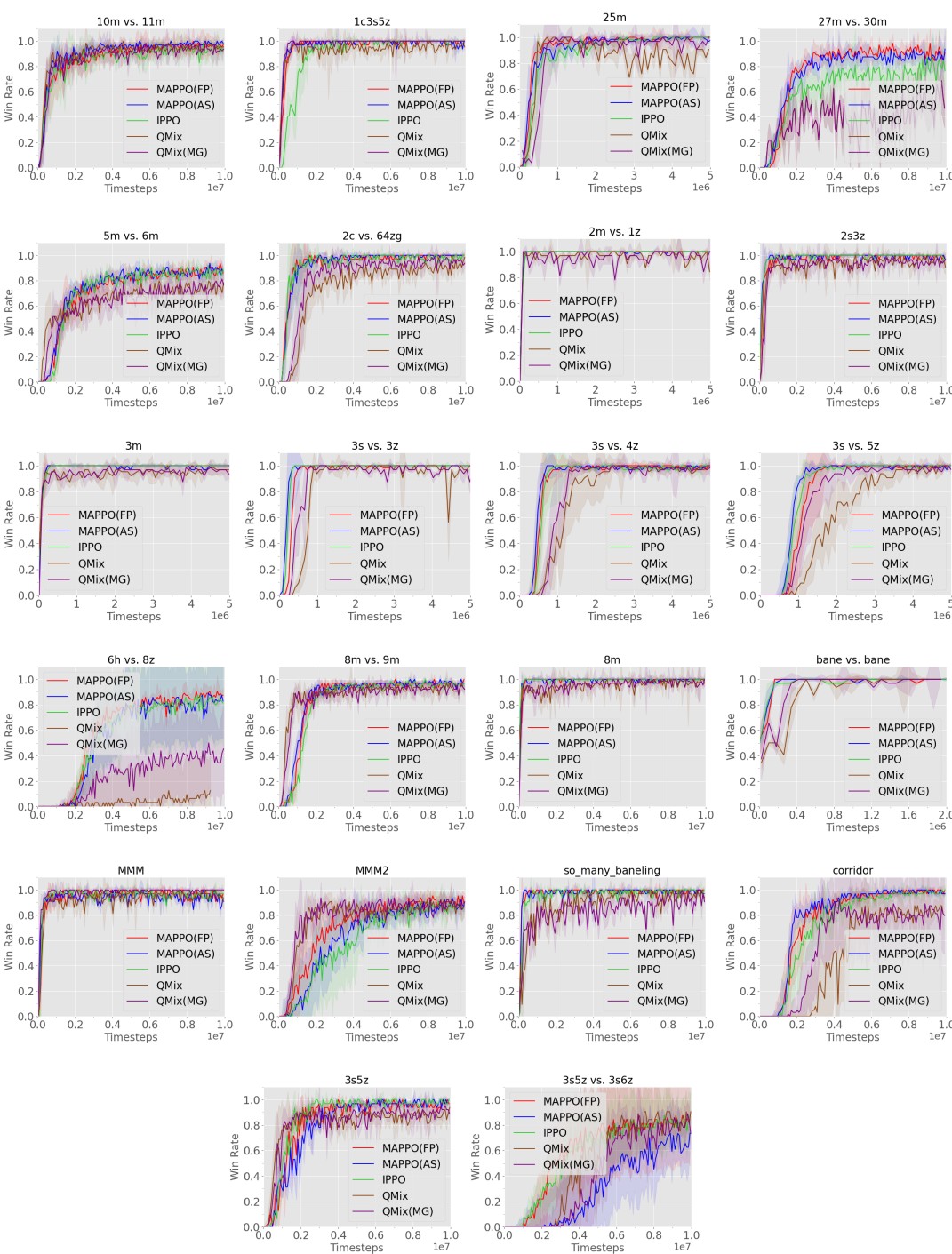

Figure 4: Median evaluation win rate of 23 maps in the SMAC domain.

| Map | Difficulty | MAPPO(FP) | MAPPO(AS) | IPPO | QMix | QMix(MG) | RODE | QPLEX | CWQMix | AIQMix |
|---|---|---|---|---|---|---|---|---|---|---|
| 2m_vs_1z | Easy | 100.0(0.0) | 100.0(0.0) | 100.0(0.0) | 95.3(5.2) | 96.9(4.5) | / | / | / | / |
| 3m | Easy | 100.0(0.0) | 100.0(1.5) | 100.0(0.0) | 96.9(1.3) | 96.9(1.7) | / | / | / | / |
| 2s_vs_1sc | Easy | 100.0(0.0) | 100.0(0.0) | 100.0(1.5) | 96.9(2.9) | 100.0(1.4) | 100(0.0) | 98.4(1.6) | 100(0.0) | 100(0.0) |
| 2s3z | Easy | 100.0(0.7) | 100.0(1.5) | 100.0(0.0) | 95.3(2.5) | 96.1(2.1) | 100(0.0) | 100(4.3) | 93.7(2.2) | 96.9(0.7) |
| 3s_vs_3z | Easy | 100.0(0.0) | 100.0(0.0) | 100.0(0.0) | 96.9(12.5) | 96.9(3.7) | / | / | / | / |
| 3s_vs_4z | Easy | 100.0(1.3) | 98.4(1.6) | 99.2(1.5) | 97.7(1.9) | 97.7(1.4) | / | / | / | / |
| so_many_baneling | Easy | 100.0(0.0) | 100.0(0.7) | 100.0(1.5) | 96.9(2.3) | 92.2(5.8) | / | / | / | / |
| 8m | Easy | 100.0(0.0) | 100.0(0.0) | 100.0(0.7) | 97.7(1.9) | 96.9(2.0) | / | / | / | / |
| MMM | Easy | 96.9(2.6) | 93.8(1.5) | 96.9(0.0) | 95.3(2.5) | 100.0(0.0) | / | / | / | / |
| 1c3s5z | Easy | 100.0(0.0) | 96.9(2.6) | 100.0(0.0) | 96.1(1.7) | 100.0(0.5) | 100(0.0) | 96.8(1.6) | 96.9(1.4) | 92.2(10.4) |
| bane_vs_bane | Easy | 100.0(0.0) | 100.0(0.0) | 100.0(0.0) | 100.0(0.9) | 100.0(2.1) | 100(46.4) | 100(2.9) | 100(0.0) | 85.9(34.7) |
| 3s_vs_5z | Hard | 100.0(0.6) | 99.2(1.4) | 100.0(0.0) | 98.4(2.4) | 98.4(1.6) | 78.9(4.2) | 98.4(1.4) | 34.4(6.5) | 82.8(10.6) |
| 2c_vs_64zg | Hard | 100.0(0.0) | 100.0(0.0) | 98.4(1.3) | 92.2(4.0) | 95.3(1.5) | 100(0.0) | 90.6(7.3) | 85.9(3.3) | 97.6(2.3) |
| 8m_vs_9m | Hard | 96.9(0.6) | 96.9(0.6) | 96.9(0.7) | 92.2(2.0) | 93.8(2.7) | / | / | / | / |
| 25m | Hard | 100.0(1.5) | 100.0(4.0) | 100.0(0.0) | 85.9(7.1) | 96.9(3.8) | / | / | / | / |
| 5m_vs_6m | Hard | 89.1(2.5) | 88.3(1.2) | 87.5(2.3) | 75.8(3.7) | 76.6(2.6) | 71.1(9.2) | 70.3(3.2) | 57.8(9.1) | 64.1(5.5) |
| 3s5z | Hard | 96.9(0.7) | 96.9(1.9) | 96.9(1.5) | 88.3(2.9) | 92.2(1.8) | 93.75(1.95) | 96.8(2.2) | 70.3(20.3) | 96.9(2.9) |
| 10m_vs_11m | Hard | 96.9(4.8) | 96.9(1.2) | 93.0(7.4) | 95.3(1.0) | 92.2(2.0) | 95.3(2.2) | 96.1(8.7) | 75.0(3.3) | 96.9(1.4) |
| MMM2 | Super Hard | 90.6(2.8) | 87.5(5.1) | 86.7(7.3) | 87.5(2.6) | 88.3(2.2) | 89.8(6.7) | 82.8(20.8) | 0.0(0.0) | 67.2(12.4) |
| 3s5z_vs_3s6z | Super Hard | 84.4(34.0) | 63.3(19.2) | 82.8(19.1) | 82.8(5.3) | 82.0(4.4) | 96.8(25.11) | 10.2(11.0) | 53.1(12.9) | 0.0(0.0) |
| 27m_vs_30m | Super Hard | 93.8(2.4) | 85.9(3.8) | 69.5(11.8) | 39.1(9.8) | 39.1(9.8) | 96.8(1.5) | 43.7(18.7) | 82.8(7.8) | 62.5(34.3) |
| 6h_vs_8z | Super Hard | 88.3(3.7) | 85.9(30.9) | 84.4(33.3) | 9.4(2.0) | 39.8(4.0) | 78.1(37.0) | 1.5(31.0) | 49.2(14.8) | 0.0(0.0) |
| corridor | Super Hard | 100.0(1.2) | 98.4(0.8) | 98.4(3.1) | 84.4(2.5) | 81.2(5.9) | 65.6(32.1) | 0.0(0.0) | 0.0(0.0) | 12.5(7.6) |

Table 2: Median evaluation win rate and standard deviation on all the SMAC maps for different methods, using at most 10M training timesteps.

| Map | Difficulty | MAPPO(FP)* | MAPPO(AS)* | IPPO* | QMix* | QMix(MG)* | RODE | QPLEX | CWQMix | AIQMix |
|---|---|---|---|---|---|---|---|---|---|---|
| 2m_vs_1z | Easy | 100.0(0.0) | 100.0(0.0) | 100.0(0.0) | 96.9(2.8) | 96.9(4.7) | / | / | / | / |
| 3m | Easy | 100.0(0.0) | 100.0(1.5) | 100.0(0.0) | 92.2(2.7) | 96.9(2.1) | / | / | / | / |
| 2s_vs_1sc | Easy | 100.0(0.0) | 100.0(0.0) | 100.0(0.0) | 96.9(1.2) | 96.9(4.6) | 100(0.0) | 98.4(1.6) | 100(0.0) | 100(0.0) |
| 2s3z | Easy | 96.9(1.5) | 96.9(1.5) | 100.0(0.0) | 95.3(3.9) | 92.2(2.3) | 100(0.0) | 100(4.3) | 93.7(2.2) | 96.9(0.7) |
| 3s_vs_3z | Easy | 100.0(0.0) | 100.0(0.0) | 100.0(0.0) | 100.0(1.5) | 100.0(1.5) | / | / | / | / |
| 3s_vs_4z | Easy | 100.0(2.1) | 100.0(1.5) | 100.0(1.4) | 87.5(3.2) | 98.4(0.8) | / | / | / | / |
| so_many_baneling | Easy | 100.0(1.5) | 96.9(1.5) | 96.9(1.5) | 81.2(7.2) | 78.1(6.7) | / | / | / | / |
| 8m | Easy | 100.0(0.0) | 100.0(0.0) | 100.0(1.5) | 93.8(5.1) | 93.8(2.7) | / | / | / | / |
| MMM | Easy | 93.8(2.6) | 96.9(1.5) | 96.9(1.5) | 95.3(3.9) | 100.0(1.2) | / | / | / | / |
| 1c3s5z | Easy | 100.0(0.0) | 96.9(2.6) | 93.8(5.1) | 95.3(1.2) | 98.4(1.4) | 100(0.0) | 96.8(1.6) | 96.9(1.4) | 92.2(10.4) |
| bane_vs_bane | Easy | 100.0(0.0) | 100.0(0.0) | 100.0(0.0) | 100.0(0.0) | 100.0(0.0) | 100(46.4) | 100(2.9) | 100(0.0) | 85.9(34.7) |
| 3s_vs_5z | Hard | 98.4(5.5) | 100.0(1.2) | 100.0(2.4) | 56.2(8.8) | 90.6(2.2) | 78.9(4.2) | 98.4(1.4) | 34.4(6.5) | 82.8(10.6) |
| 2c_vs_64zg | Hard | 96.9(3.1) | 95.3(3.5) | 93.8(9.2) | 70.3(3.8) | 84.4(3.7) | 100(0.0) | 90.6(7.3) | 85.9(3.3) | 97.6(2.3) |
| 8m_vs_9m | Hard | 84.4(5.1) | 87.5(2.1) | 76.6(5.6) | 85.9(2.9) | 85.9(4.7) | / | / | / | / |
| 25m | Hard | 96.9(3.1) | 93.8(2.9) | 93.8(5.0) | 96.9(4.0) | 93.8(5.7) | / | / | / | / |
| 5m_vs_6m | Hard | 65.6(14.1) | 68.8(8.2) | 64.1(7.7) | 54.7(3.5) | 56.2(2.1) | 71.1(9.2) | 70.3(3.2) | 57.8(9.1) | 64.1(5.5) |
| 3s5z | Hard | 71.9(11.8) | 53.1(15.4) | 84.4(12.1) | 85.9(4.6) | 89.1(2.6) | 93.75(1.95) | 96.8(2.2) | 70.3(20.3) | 96.9(2.9) |
| 10m_vs_11m | Hard | 81.2(8.3) | 89.1(5.5) | 87.5(17.5) | 82.8(4.1) | 85.9(2.3) | 95.3(2.2) | 96.1(8.7) | 75.0(3.3) | 96.9(1.4) |
| MMM2 | Super Hard | 51.6(21.9) | 28.1(29.6) | 26.6(27.8) | 82.8(4.0) | 79.7(3.4) | 89.8(6.7) | 82.8(20.8) | 0.0(0.0) | 67.2(12.4) |
| 3s5z_vs_3s6z | Super Hard | 75.0(36.3) | 18.8(37.4) | 65.6(25.9) | 56.2(11.3) | 39.1(4.7) | 96.8(25.11) | 10.2(11.0) | 53.1(12.9) | 0.0(0.0) |
| 27m_vs_30m | Super Hard | 93.8(3.8) | 89.1(6.5) | 73.4(11.5) | 34.4(5.4) | 34.4(5.4) | 96.8(1.5) | 43.7(18.7) | 82.8(7.8) | 62.5(34.3) |
| 6h_vs_8z | Super Hard | 78.1(5.6) | 81.2(31.8) | 78.1(33.1) | 3.1(1.5) | 29.7(6.3) | 78.1(37.0) | 1.5(31.0) | 49.2(14.8) | 0.0(0.0) |
| corridor | Super Hard | 93.8(3.5) | 93.8(2.8) | 89.1(9.1) | 64.1(14.3) | 81.2(1.5) | 65.6(32.1) | 0.0(0.0) | 0.0(0.0) | 12.5(7.6) |

Table 3: Median evaluation win rate and standard deviation on all the SMAC maps for different methods, Columns with "*" display results using the same number of timesteps as RODE.

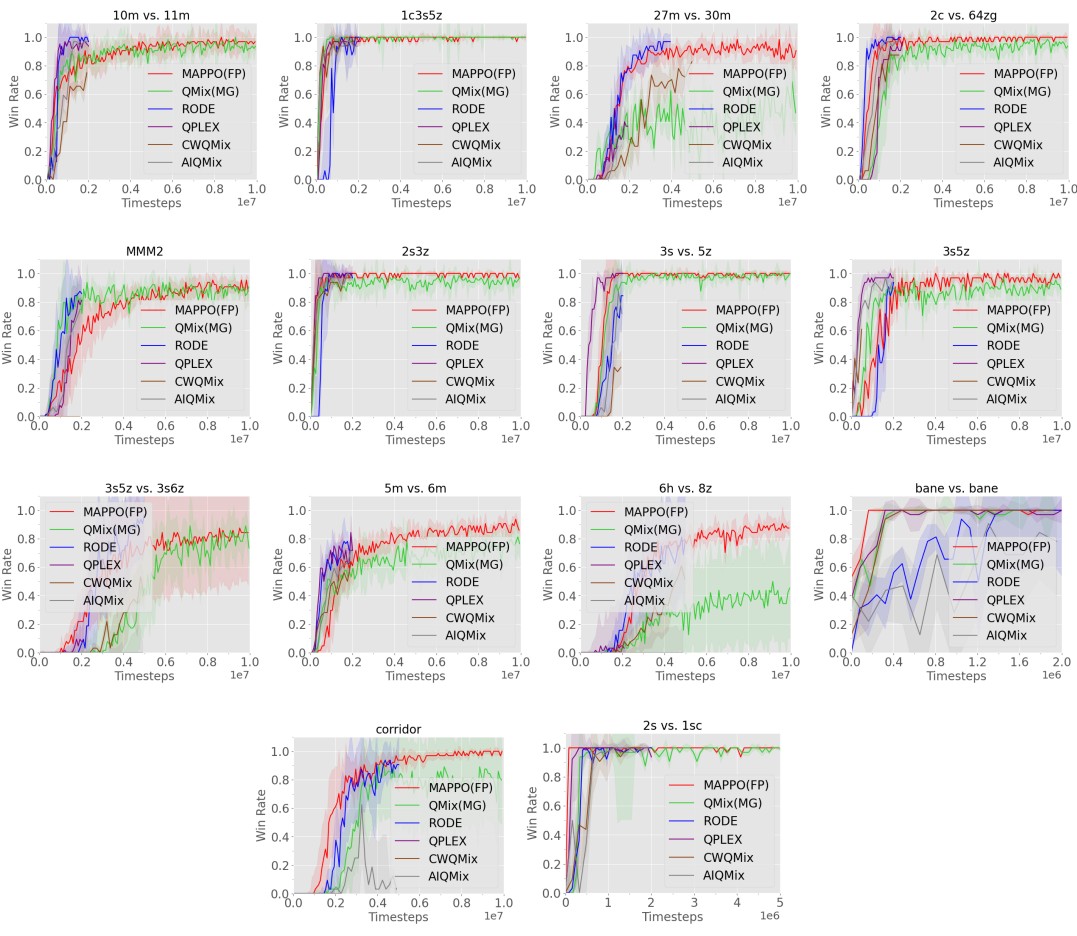

Figure 5: Median evaluation win rate of MAPPO(FP), QMix(MG), RODE, QPlEX, CWQMix and AIQMix in the SMAC domain.

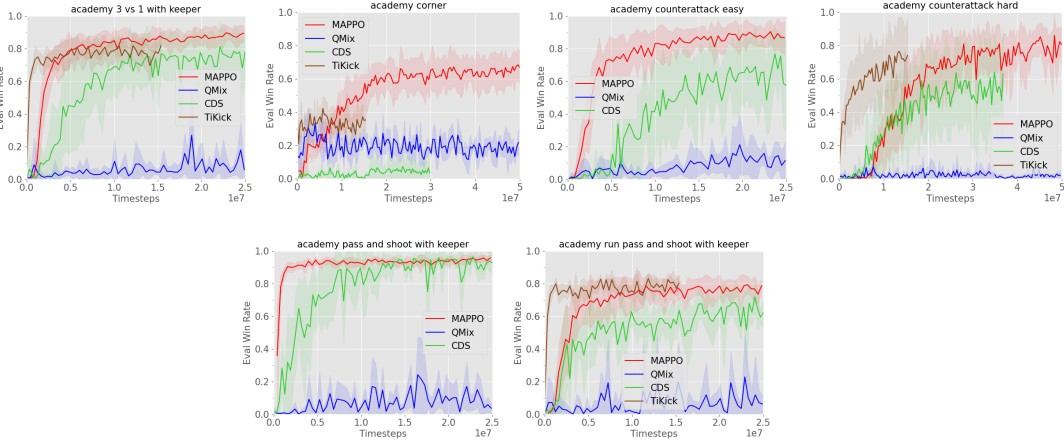

Figure 6: Mean evaluation win rate of MAPPO, QMix, CDS, TiKick in the GRF domain.

| common hyperparameters | value |
|---|---|
| recurrent data chunk length | 10 |
| gradient clip norm | 10.0 |
| gae lamda | 0.95 |
| gamma | 0.99 |
| value loss | huber loss |
| huber delta | 10.0 |
| batch size | num envs $\times$ buffer length $\times$ num agents |
| mini batch size | batch size / mini-batch |
| optimizer | Adam |
| optimizer epsilon | 1e-5 |
| weight decay | 0 |
| network initialization | Orthogonal |
| use reward normalization | True |
| use feature normalization | True |

Table 4: Common hyperparameters used in MAPPO across all domains.

| common hyperparameters | value |
|---|---|
| gradient clip norm | 10.0 |
| random episodes | 5 |
| epsilon | $1.0 \rightarrow 0.05$ |
| epsilon anneal time | 50000 timesteps |
| train interval | 1 episode |
| gamma | 0.99 |
| critic loss | mse loss |
| buffer size | 5000 episodes |
| batch size | 32 episodes |
| optimizer | Adam |
| optimizer eps | 1e-5 |
| weight decay | 0 |
| network initialization | Orthogonal |
| use reward normalization | True |
| use feature normalization | True |

Table 5: Common hyperparameters used in QMix and MADDPG across all domains.

| hyperparameters | value |
|---|---|
| num envs | MAPPO: 128 |
| buffer length | MAPPO: 25 |
| num GRU layers | 1 |
| RNN hidden state dim | 64 |
| fc layer dim | 64 |
| num fc | 2 |
| num fc after | 1 |

Table 6: Common hyperparameters used in the MPE domain for MAPPO, MADDPG, and QMix.

| hyperparameters | value |
|---|---|
| num envs | MAPPO:8 |
| buffer length | MAPPO: 400 |
| num GRU layers | 1 |
| RNN hidden state dim | 64 |
| fc layer dim | 64 |
| num fc | 2 |
| num fc after | 1 |

Table 7: Common hyperparameters used in the SMAC domain for MAPPO and QMix.

| hyperparameters | value |
|---|---|
| num envs | 1000 |
| buffer length | 100 |
| fc layer dim | 512 |
| num fc | 2 |

Table 8: Common hyperparameters used in the Hanabi domain for MAPPO.

| hyperparameters | value |
|---|---|
| parallel envs | MAPPO: 50 QMix: 1 |
| horizon length | 199 |
| num GRU layers | 1 |
| RNN hidden state dim | 64 |
| fc layer dim | 64 |
| num fc | 2 |
| num fc after | 1 |

Table 9: Common hyperparameters used in the GRF domain for MAPPO and QMix.

| Domains | lr | epoch | mini-batch | activation | clip | gain | entropy coef | network |
|---|---|---|---|---|---|---|---|---|
| MPE | [1e-4,5e-4,7e-4,1e-3] | [5,10,15,20] | [1,2,4] | [ReLU,Tanh] | [0.05,0.1,0.15,0.2,0.3,0.5] | [0.01,1] | / | [mlp,rnn] |
| SMAC | [1e-4,5e-4,7e-4,1e-3] | [5,10,15] | [1,2,4] | [ReLU,Tanh] | [0.05,0.1,0.15,0.2,0.3,0.5] | [0.01,1] | / | [mlp,rnn] |
| Hanabi | [1e-4,5e-4,7e-4,1e-3] | [5,10,15] | [1,2,4] | [ReLU,Tanh] | [0.05,0.1,0.15,0.2,0.3,0.5] | [0.01,1] | [0.01, 0.015] | [mlp,rnn] |
| Football | [1e-4,5e-4,7e-4,1e-3] | [5,10,15] | [1,2,4] | [ReLU,Tanh] | [0.01,1] | [0.01, 0.015] | [mlp,rnn] | |

Table 10: Sweeping procedure of MAPPO cross all domains.

| Domains | lr | tau | hard interval | activation | gain |
|---|---|---|---|---|---|
| MPE | [1e-4,5e-4,7e-4,1e-3] | [0.001,0.005,0.01] | [100,200,500] | [ReLU,Tanh] | [0.01,1] |
| SMAC | [1e-4,5e-4,7e-4,1e-3] | [0.001,0.005,0.01] | [100,200,500] | [ReLU,Tanh] | [0.01,1] |

Table 11: Sweeping procedure of QMix in the MPE and SMAC domains.

| Domains | lr | tau | activation | gain | network |
|---|---|---|---|---|---|
| MPE | [1e-4,5e-4,7e-4,1e-3] | [0.001,0.005,0.01] | [ReLU,Tanh] | [0.01,1] | [mlp,rnn] |

Table 12: Sweeping procedure of MADDPG in the MPE domain.

| Scenarios | lr | gain | network | MAPPO | | | | MADDPG | | QMix | |
|---|---|---|---|---|---|---|---|---|---|---|---|
| | | | | epoch | mini-batch | activation | tau | activation | tau | hard interval | activation |
| Spread | 7e-4 | 0.01 | rnn | 10 | 1 | Tanh | 0.005 | ReLU | / | 100 | ReLU |
| Reference | 7e-4 | 0.01 | rnn | 15 | 1 | ReLU | 0.005 | ReLU | 0.005 | / | ReLU |
| Comm | 7e-4 | 0.01 | rnn | 15 | 1 | Tanh | 0.005 | ReLU | 0.005 | / | ReLU |

Table 13: Adopted hyperparameters used for MAPPO, MADDPG and QMix in the MPE domain.

| Maps | lr | activation | MAPPO | | | | | | | QMix | |
|---|---|---|---|---|---|---|---|---|---|---|---|
| | | | epoch | mini-batch | clip | gain | network | stacked frames | | hard interval | gain |
| 2m vs. 1z | 5e-4 | ReLU | 15 | 1 | 0.2 | 0.01 | rnn | 1 | | 200 | 0.01 |
| 3m | 5e-4 | ReLU | 15 | 1 | 0.2 | 0.01 | rnn | 1 | | 200 | 0.01 |
| 2s vs. 1sc | 5e-4 | ReLU | 15 | 1 | 0.2 | 0.01 | rnn | 1 | | 200 | 0.01 |
| 3s vs. 3z | 5e-4 | ReLU | 15 | 1 | 0.2 | 0.01 | rnn | 1 | | 200 | 0.01 |
| 3s vs. 4z | 5e-4 | ReLU | 15 | 1 | 0.2 | 0.01 | mlp | 4 | | 200 | 0.01 |
| 3s vs. 5z | 5e-4 | ReLU | 15 | 1 | 0.05 | 0.01 | mlp | 4 | | 200 | 0.01 |
| 2c vs. 64zg | 5e-4 | ReLU | 5 | 1 | 0.2 | 0.01 | rnn | 1 | | 200 | 0.01 |
| so many baneling | 5e-4 | ReLU | 15 | 1 | 0.2 | 0.01 | rnn | 1 | | 200 | 0.01 |
| 8m | 5e-4 | ReLU | 15 | 1 | 0.2 | 0.01 | rnn | 1 | | 200 | 0.01 |
| MMM | 5e-4 | ReLU | 15 | 1 | 0.2 | 0.01 | rnn | 1 | | 200 | 0.01 |
| 1c3s5z | 5e-4 | ReLU | 15 | 1 | 0.2 | 0.01 | rnn | 1 | | 200 | 0.01 |
| 8m vs. 9m | 5e-4 | ReLU | 15 | 1 | 0.05 | 0.01 | rnn | 1 | | 200 | 0.01 |
| bane vs. bane | 5e-4 | ReLU | 15 | 1 | 0.2 | 0.01 | rnn | 1 | | 200 | 0.01 |
| 25m | 5e-4 | ReLU | 10 | 1 | 0.2 | 0.01 | rnn | 1 | | 200 | 0.01 |
| 5m vs. 6m | 5e-4 | ReLU | 10 | 1 | 0.05 | 0.01 | rnn | 1 | | 200 | 0.01 |
| 3s5z | 5e-4 | ReLU | 5 | 1 | 0.2 | 0.01 | rnn | 1 | | 200 | 0.01 |
| MMM2 | 5e-4 | ReLU | 5 | 2 | 0.2 | 1 | rnn | 1 | | 200 | 0.01 |
| 10m vs. 11m | 5e-4 | ReLU | 10 | 1 | 0.2 | 0.01 | rnn | 1 | | 200 | 0.01 |
| 3s5z vs. 3s6z | 5e-4 | ReLU | 5 | 1 | 0.2 | 0.01 | rnn | 1 | | 200 | 1 |
| 27m vs. 30m | 5e-4 | ReLU | 5 | 1 | 0.2 | 0.01 | rnn | 1 | | 200 | 1 |
| 6h vs. 8z | 5e-4 | ReLU | 5 | 1 | 0.2 | 0.01 | mlp | 1 | | 200 | 1 |
| corridor | 5e-4 | ReLU | 5 | 1 | 0.2 | 0.01 | mlp | 1 | | 200 | 1 |

Table 14: Adopted hyperparameters used for MAPPO and QMix in the SMAC domain.

| Tasks | MAPPO | | | | | | | |
|---|---|---|---|---|---|---|---|---|
| | lr | epoch | mini-batch | activation | gain | entropy coef | network |
| 2-player | actor:7e-4 critic:1e-3 | 15 | 1 | ReLU | 0.01 | 0.015 | mlp |

Table 15: Adopted hyperparameters used for MAPPO in the Hanabi domain.

| Scenarios | lr | activation | buffer length | MAPPO | | | | QMix | |
|---|---|---|---|---|---|---|---|---|---|
| | | | | epoch | mini-batch | gain | network | hard interval | gain |
| 3v.1 | 5e-4 | ReLU | 200 | 15 | 2 | 0.01 | rnn | 200 | 0.01 |
| Corner | 5e-4 | ReLU | 1000 | 15 | 2 | 0.01 | rnn | 200 | 0.01 |
| CA(easy) | 5e-4 | ReLU | 200 | 15 | 2 | 0.01 | rnn | 200 | 0.01 |
| CA(hard) | 5e-4 | ReLU | 1000 | 15 | 2 | 0.01 | rnn | 200 | 0.01 |
| PS | 5e-4 | ReLU | 200 | 15 | 2 | 0.01 | rnn | 200 | 0.01 |
| RPS | 5e-4 | ReLU | 200 | 15 | 2 | 0.01 | rnn | 200 | 0.01 |

Table 16: Adopted hyperparameters used for MAPPO and QMix in the Football domain.

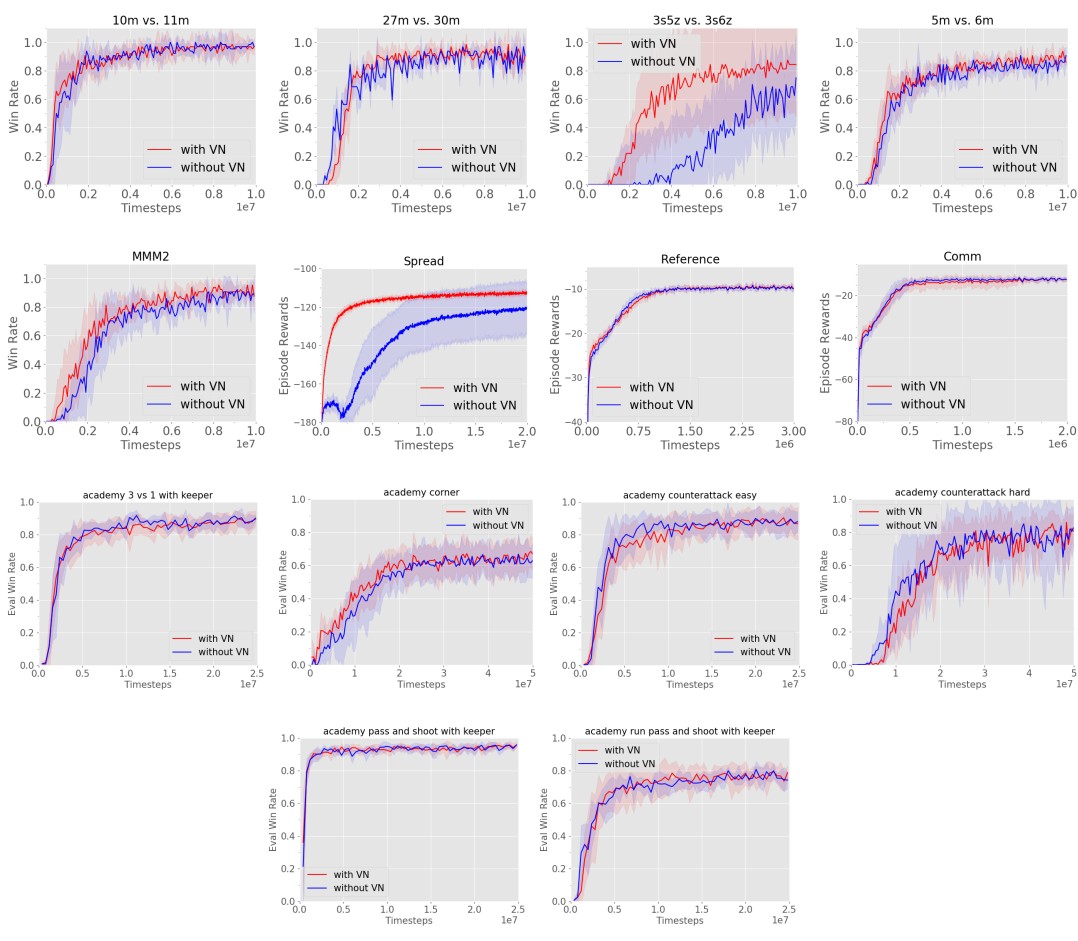

Figure 7: Ablation studies demonstrating the effect of Value Normalization(VN) on MAPPO's performance in the MPE, SMAC, and GRF domains.

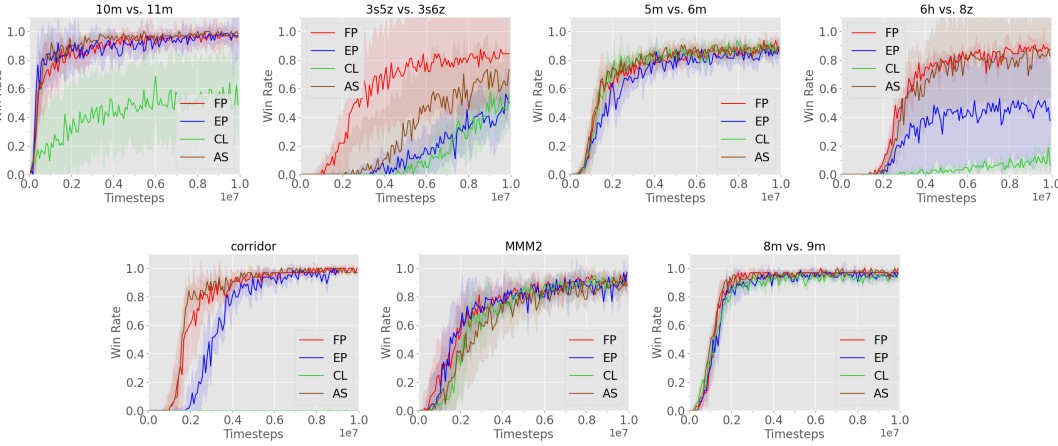

Figure 8: Ablation studies demonstrating the effect of different global state on MAPPO's performance in the SMAC domain.

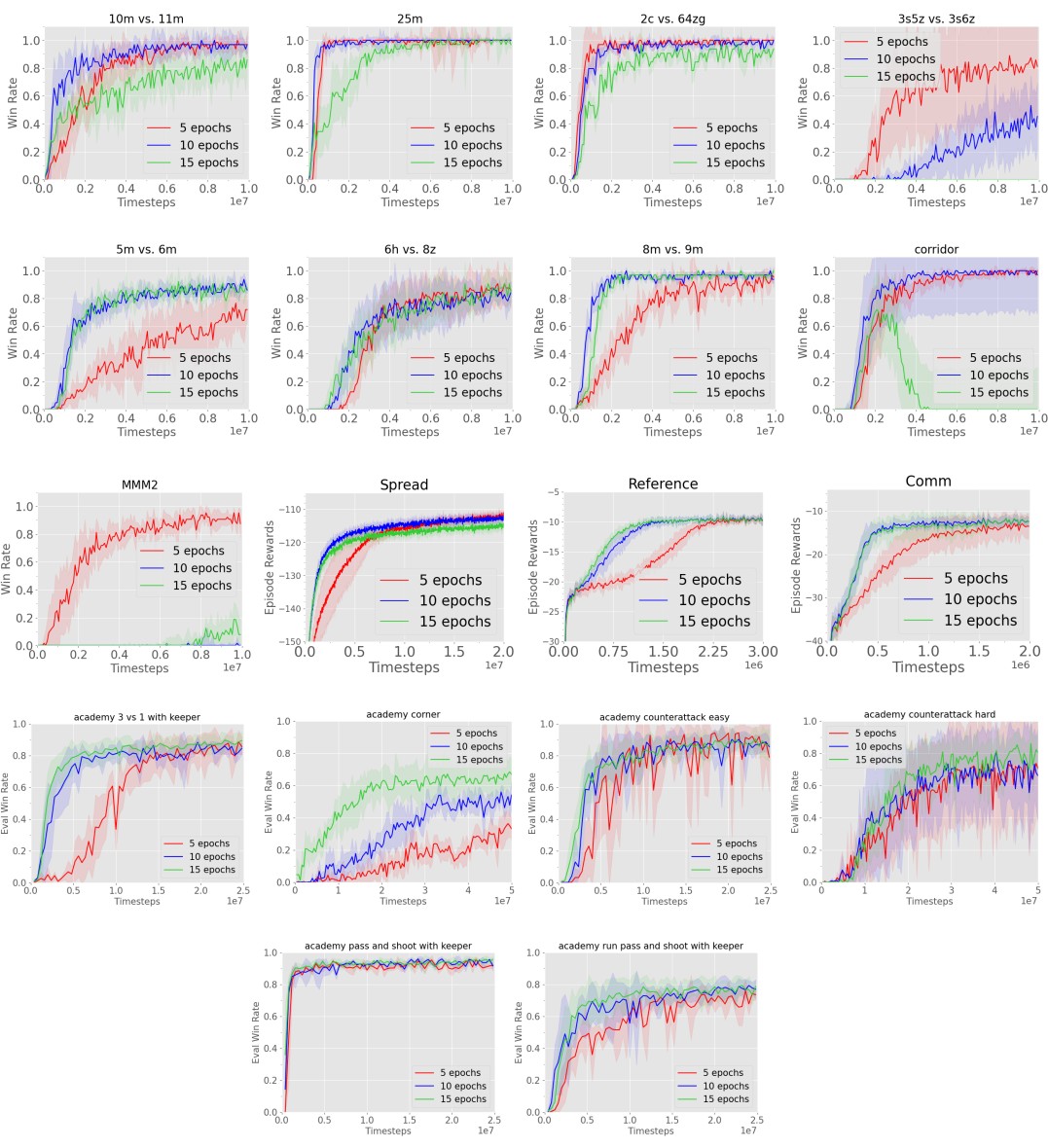

Figure 9: Ablation studies demonstrating the effect of training epochs on MAPPO's performance in the MPE, SMAC, and GRF domains.

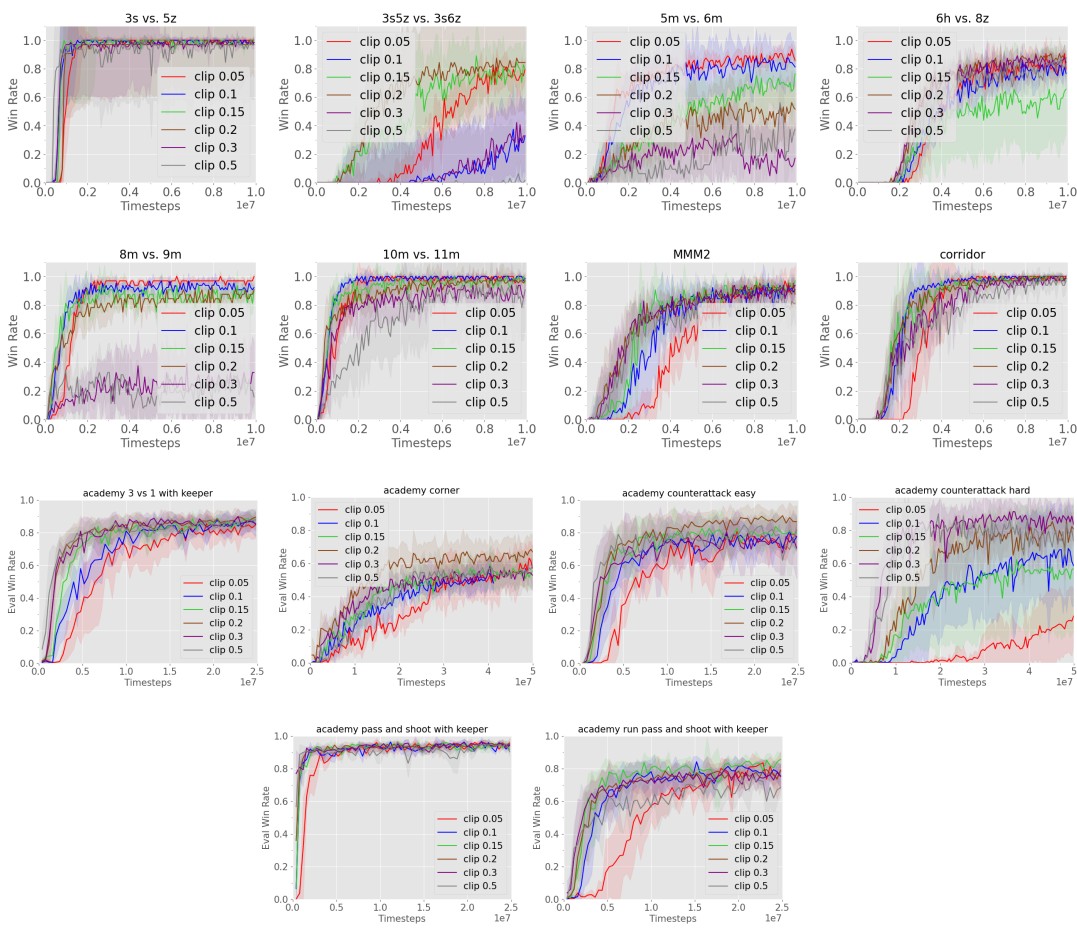

Figure 10: Ablation studies demonstrating the effect of clip term on MAPPO's performance in the SMAC and GRF domain.

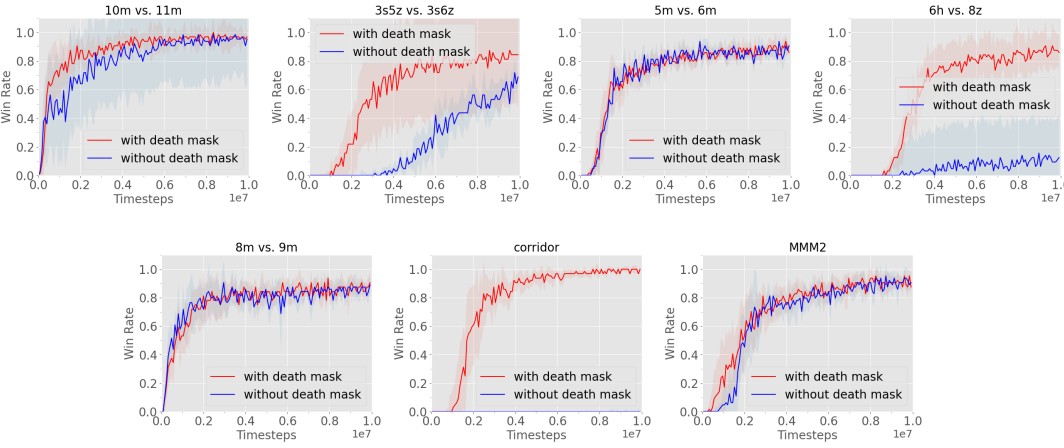

Figure 11: Ablation studies demonstrating the effect of death mask on MAPPO(FP)'s performance in the SMAC doamin.

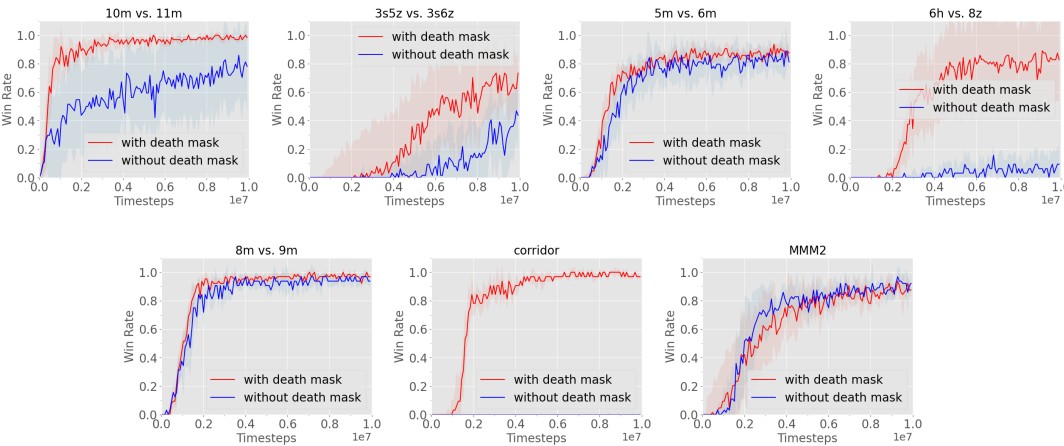

Figure 12: Ablation studies demonstrating the effect of death mask on MAPPO(AS)'s performance in the SMAC domain.

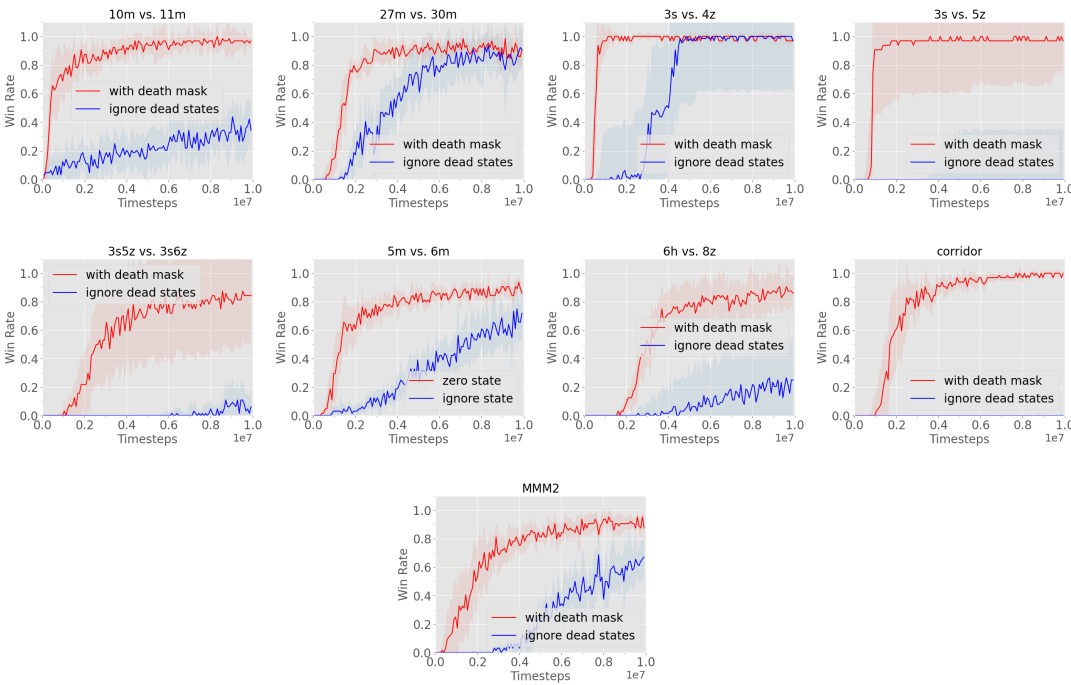

Figure 13: Ablation studies demonstrating the effect of death mask on MAPPO's performance in the SMAC domain.

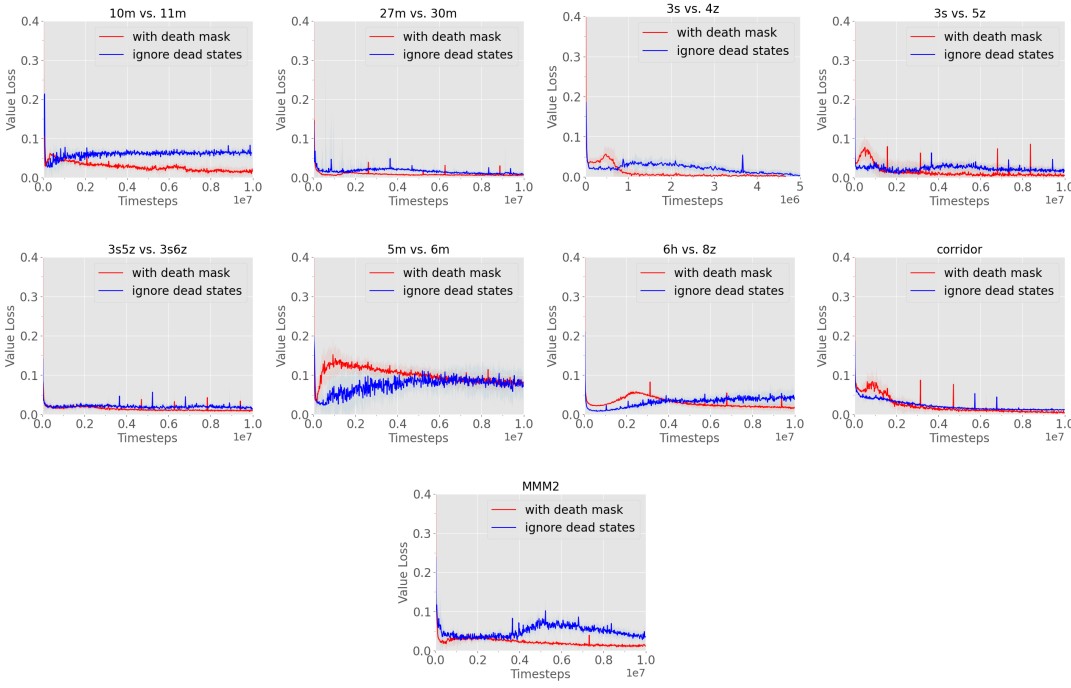

Figure 14: Effect of death mask on MAPPO's value loss in the SMAC domain.

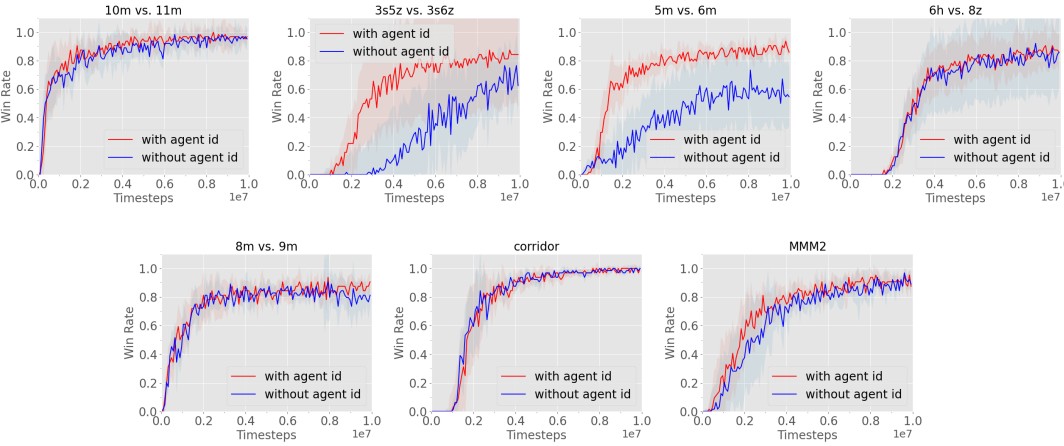

Figure 15: Ablation studies demonstrating the effect of agent id on MAPPO's performance in the SMAC domain.