# OpenReview forum: "The Surprising Effectiveness of PPO in Cooperative Multi-Agent Games"
_NeurIPS.cc/2022/Track/Datasets_and_Benchmarks — NeurIPS 2022 Datasets and Benchmarks _

### Official Review · Reviewer_Ryeg · 2022-07-07
**A comprehensive study on the performance of the multi-agent PPO variant in the multi-agent settings**

**Rating:** 6
**Confidence:** 4

**Strengths:**

**(+)** Extensive experiments to prove the authors' points and the experiments are carefully planned and laid out in a thoughtful way.
\
\
**(+)** Well-structured main text, well-presented figures and tables. Empirical analysis and suggestions are provided for each influential factor. Most of the important details regarding the experiment setup are there.
\
\
**(+)** I generally agree with the authors' finding of "critical point" batch size. Some literature I have read in the past has reported a similar conclusion.

**Weaknesses:**

Note: I use **(~)** to denote my doubt or question but may not necessarily be a weakness of the paper.

**(-)** I'm worried that this work becomes a technical guide on how to tune hypermeters of the MAPPO algorithm while training in the SMAC environment. Though I definitely appreciate most of the authors' suggestions drawn from their experimental analysis since SMAC has become one of the most standard benchmarks for the MARL algorithms. I was wondering if the authors believe that to what extent their conclusions may be applicable to other MARL environments, where data frequency, reward landscape, the difficulty of credit assignment, etc., will differ. For instance, have you thought about performing some benchmarks on temporal sequential social dilemma (SSD) environments (i.e. involve mixed-motive agents) to see if your key findings hold across the board? I'm raising this concern because Section 5.2 seems specifically applies to the information structure of SMAC.
\
\
**(~)** Perhaps blaming everything on the non-stationarity of MARL might be too generic? One example might be Line 266-268. I am hoping for more insightful reasoning behind the performance gap caused by a change in hyperparameter.
\
\
**(-)** I believe that many factors in the author's analysis, such as PPO batch size, number of mini-batches, training epochs, and rollout fragment length, are tightly entangled together. Therefore, I think the author should combine Section 5.3 and 5.5 and perhaps add the rollout fragment length since they would affect the GAE horizon. Maybe more extensive grid searches that test out different combinations of these hyperparameters I mentioned in the first sentence could better convince me.
\
\
**(-)** From what I can see in Tables 3 and 4 from the supplementary material, QMix and MADDPG employ $\epsilon$\_greedy exploration while it is not clear to me what the PPO has used. Stochastic sampling?
\
\
**(-)** I have mixed opinions regarding the use of the "Median evaluation win rate" metric for SMAC. It makes some of the comparisons inconclusive especially when 100% win rates are obtainable. What about showing the average return?

**Additional Feedback:**

* Reference [24], G. Papoudakis et. al 2020, was an accepted paper of this track last year, please update the citation accordingly.
* Some tables and figures have gone over the top margin limit for almost an inch. Clear signs of abusing reduce `\vspace` command and it shows when Line number 259 and 260 are almost squished together. The titles, the figure and the text are all crammed together and causing minor discomfort while I was reviewing this paper.

This work experienced a few iterations of modifications over the past year, and I will admit that it is now in better shape compared to the previous versions. Therefore I am voting for marginal acceptance of this work.

**Clarity:**

The figures and tables are information and well-structured. The main text is well-written and easy to follow.

**Correctness:**

The evaluation method looks correct and the results seem reliable. Most experiments that I am aware of were run on multiple seeds (3~6). Training curves were provided with confidence intervals. Error bars and one standard deviation were provided in the bar charts and the tables respectively.

Line 152:
> "take the median of the final ten evaluations as the performance for each seed"

Are they taking the top-10 evaluations sorted by win-lose or are you taking the actual last 10 games from the sequence of evaluations? Why not take the median over all 32 games?

Some data in Table 1 indicate significantly large variances in win rates, for example, the results for MAPPO(AS), IPPO in 6hvs8z, and MAPPO(FP) in 3s5zvs3s6z.

**Documentation:**

The GitHub repos have been made public for a while now (about 1 year) and are fairly well maintained. There are a dedicated website and a blog post written about the project. Most Github issues and pull requests are resolved in a timely manner. Instructions to reproduce the results are provided on the GitHub page.

**Ethics:**

Briefly discussed in the conclusion. No issues here.

**Relation To Prior Work:**

The authors have cited most of the relevant influential works that I can think of (namely [9],[1],[24]) and differ from them by analyzing the performance of the PPO algorithms exclusively in multi-agent environments or have performed more experiments on SMAC.

**Summary And Contributions:**

Most previous works regarding extensive benchmarks of the PPO algorithm focused on the single-agent setting, hence this work is different in that all of the benchmarks performed in this work concern the multi-agent setting.
First, the authors compare the performance of MAPPO against its variants and other MARL algorithms in four cooperative game environments. Then the authors proceed to identify five key implementation and hyperparameter factors that are influential in PPO's performance in multi-agent settings, with a particular focus on the SMAC environment. The authors provide their empirical analysis and suggestions for each of the factors accordingly. To my knowledge, this paper is one of the early works that systematically study the performance of MAPPO.
\
Compared to the version published on ArVix, the death mask section is moved to the Supplementary Material while a new section regarding PPO batch size is made to replace it.

---

> ### Author Response · Authors · 2022-08-13
> **Thank you for the feedback. Here are our responses:**
>
> We thank the reviewer for the review and feedback. Here are our responses:
> - Regarding the reviewer’s concern that the suggestions are specific to SMAC - the majority of experiments involve SMAC because SMAC offers the largest number of environments in which to study the impact of various aspects of PPO. However, in Section 5, we provide results from the MPEs (Figures 2, 6) and GRF (Figure 8). Furthermore, all of the suggestions we give are applied wherever possible in every environment. For instance, in Hanabi, we utilize all suggestions, and in GRF, we utilize all suggestions with the exception of the suggestion regarding value function input representations (Sec. 5.2) since GRF is a fully observed environment. The hyperparameter table in Appendix C.3 also demonstrates that we have applied our suggestions in all benchmark environments. We are running additional ablation studies on the GRF domain, which takes some time, and promise to include new plots in the appendix when they are finished. We do not study environments such as the SSDs since our work focuses on **cooperative** settings as detailed in the title and introduction.
> - We acknowledge that attributing non-stationarity as the cause for the patterns observed in Sections 5.3 and 5.4 is a high-level hypothesis that is based on the observed experimental results. We believe that further theoretical analysis is required to investigate this hypothesis further; however, the focus of our work is an empirical analysis of PPO in cooperative multi-agent settings and hence we leave theoretical analysis to future work.
> - We use the maximum game length as the rollout fragment length in all domains. This guarantees enough length for proper computation of the GAE.
> While it is true that hyperparameter settings affect one another, we performed the largest possible grid search given our computational resources - this grid search is fully detailed in the appendix. We also note that the conclusions we draw based on this grid search in Sec. 5 hold consistently across our testing domains.
> - We utilize the Median evaluation win rate because the majority of past works we compare to utilize this metric [1, 2, 3], and we aim to establish a fair comparison to these methods.
>
>   To clarify the computation of our success rates, throughout the training process, we evaluate over 32 games and compute the success rate by averaging over those 32 games.To compute the final success rate for the seed, we take the success rates over the final 10 evaluations during training and compute the median of those values. The final reported value is the median of each seed’s success rate.
> - PPO utilizes stochastic sampling as the exploration strategy.
> - We also acknowledge that several experiments in SMAC have high variances (this is true for both PPO experiments and baselines such as RODE) - hence, we bold all values within 1 std of the maximum for each map.
>
> **References:**
>
> [1] Tabish Rashid, Mikayel Samvelyan, Christian Schroeder, Gregory Farquhar, Jakob Foerster, and Shimon Whiteson. QMIX: Monotonic value function factorisation for deep multi-agent reinforcement learning. In Jennifer Dy and Andreas Krause, editors, Proceedings of the 35th International Conference on Machine Learning, volume 80 of Proceedings of Machine Learning Research, pages 4295–4304. PMLR, 10–15 Jul 2018.
>
> [2] Tonghan Wang, Tarun Gupta, Anuj Mahajan, Bei Peng, Shimon Whiteson, and Chongjie Zhang. RODE: Learning roles to decompose multi-agent tasks. In International Conference on Learning Representations, 2021.
>
> [3] Jianhao Wang, Zhizhou Ren, Terry Liu, Yang Yu, and Chongjie Zhang. {QPLEX}: Duplex dueling multi-agent q-learning. In International Conference on Learning Representations, 2021.

---

> > ### Comment · Reviewer_Ryeg · 2022-08-24
> > **Feedback after rebutall**
> >
> > I appreciate your time and effort in this rebuttal. It has addressed some of my concerns. Reviewer Xc16 raised a very good point regarding the parameter-sharing setting for your evaluations, and I see that the authors had performed additional experiments to cover for it. However, as the other two reviewers pointed out, the manuscript will need some serious re-organizations, and I still have concerns over the paper's specificity to SMAC though it is common knowledge that RL algorithms are sensitive to hyperparameter settings.
> >
> > Given the influence of this paper in the community (as a few of my colleagues using the repo themselves) and the amount of evaluations presented in the paper, I am leaning toward accepting this work to this track.

---

> > > ### Author Response · Authors · 2022-08-27
> > > **We have reorganized the main paper and updated the GRF results in the appendix with changes in red.**
> > >
> > > We thank the reviewer for the valuable feedback and review. We have reorganized the main paper according to all reviewers' suggestions. The detailed changes can be found in [the shared reply](https://openreview.net/forum?id=YVXaxB6L2Pl&noteId=hsqRTXoysoW). Besides, in the appendix, we have updated the additional GRF ablation studies results in Fig. 7, 9, 10. As the figures show, the conclusion is consistent with our proposed suggestions.

---

### Official Review · Reviewer_F2xg · 2022-07-21
**OK but not good enough**

**Rating:** 5
**Confidence:** 4
**Clarity:** 1. This paper proposes MAPPO and IPPO…

**Strengths:**

As described in the paper, there is little work studying on-policy algorithm in multi-agent RL literature. This work shows the performance of PPO in four multi-agent environments.

**Weaknesses:**

Since this is a benckmark paper, my main concern is the missing of some technique details and the experiment setup, see Correctness and Clarity.

**Additional Feedback:**

N/A

**Correctness:**

+ In Section 3, the reward is conditioned on joint actions, i.e. $R(s,A)$, while the reward in objective function $J(\theta)$ is $R(s^t,a^t)$, which seems not conditioned on joint actions.

+ The experiment results in SMAC are provided in Appendix D.1. MAPPO uses AS and FP inputs, while a variant of QMIX uses MG inputs, which is “concatenation of the default environment global state”. For RODE, QPLEX, CWQMix, AIQMix, the input is not described. If different methods use different inputs, I doubt whether the comparison is reasonable. Same problem raises in other experiments, e.g., in Section 4.3, it seems the features used by MAPPO and SAD are different.



**Documentation:**

Github url is provided. The detailed instructions are provided in the repo.

**Relation To Prior Work:**

The paper makes good connections to many prior works.

**Summary And Contributions:**

This paper aims to demonstrate the promising performance of PPO in multi-agent reinforcement learning (MARL) problems compared with several off-policy MARL algorithms. The experiments are conducted in four benchmark environments, including multi-agent particle world env, StarCraft, Hanabi, and Google Research Football. Many up to date off-policy algorithms are included in the experiments and the parameter settings are given.

---

> ### Author Response · Authors · 2022-08-13
> **Thank you for the feedback, we have updated our paper accordingly.**
>
> We thank the reviewer for their feedback. We have updated our paper to clarify several of the concerns that you raise. Specifically, we have fixed the typo in the reward in the objective function and **have added text clarifying the input representations to the different baseline algorithms**.
>
> - In Table 1 (the primary table with SMAC results), all QMIX and RODE results utilize both the agent-agnostic global state and agent-specific local observations as input. Specifically, for agent $i$, the local Q-network (which computes actions at execution) takes in only the local agent-specific observation $o_i$ as input while the global mixer network takes in the agent-agnostic global state $s$ as input.  This is also the case for the other value-decomposition methods presented in Appendix Table 1 (QPLEX, CWQMix, and AIQMix).
>
>   In comparison, MAPPO also utilizes both the local agent-specific observations and the global agent-agnostic state - the actor networks utilize only the local observation $o_i$, whereas the critic network utilizes both the local observation $o_i$, and the global state $s$. In the AS setting, the critic input is created simply by concatenating $o_i$ and $s$; the FP setting merely prunes the overlapping features. Hence, **MAPPO (both AS and FP) and the value-decomposition baselines utilize the exact same set of features**.
>
>   To create an additional point of comparison, we evaluate a version of QMix, which we denote as QMix (MG)  in which the mixer network takes both the global agent-agnostic state as well as a concatenation of all agents’ local observations: $(s + o_1 + … + o_n)$. Again, QMix (MG) overall uses the same information as all the other methods. The only difference is that we now give the mixer network all available information. We remark that the performance of QMix (MG) remains relatively worse than the PPO methods.
>
>   SAD and MAPPO cannot use the same input features primarily due to differences in the structure of the algorithms. SAD learns a joint-Q function via VDN, which by design only utilizes local agent-specific observations (since there are only local agent Q-networks and no global mixer network). MAPPO, on the other hand, can leverage the fact that the critic is only used at training time to incorporate global information into the critic input. However, by design, SAD leverages additional information of greedy actions of other players in the past time steps, which is NOT used by MAPPO since MAPPO **only** operates with environment-provided information.
>
> - Regarding the differences between IPPO and MAPPO, the **only** difference is in the input to the critic - in IPPO, the critic takes as input only an agent’s local observation whereas in MAPPO, the critic takes as input global information as well as the agent’s local observation.
>
> - Regarding your final question about how the AS state can be used to train a centralized critic - this is possible because in a given trajectory, there are samples from multiple agents. For a trajectory corresponding to agent i, the critic takes as input a vector specific to agent i . For MAPPO, the critic is “centralized” in the sense that it takes in global information as well as the agent’s local observation. We also want to clarify that although all agents share the critic parameters, each agent’s critic input can be different. Hence, even though the critic is ``centralized’’ with global information, in practice, it is possible that different agents obtain different critic output values due to different AS features. Our experiments suggest that such a critic design can be empirically beneficial.

---

> > ### Comment · Reviewer_F2xg · 2022-08-19
> > **Need futher polishment**
> >
> > The main algorithm part of the paper is Sec 3, which is pretty short in the current version. Readers can not get the full image of what the authors have done to modify the PPO for multi-agent settings. The authors aim to provide a lot of information in the paper but is not well structured, for instance, a lot of MARL algorithms are compared in different experiment subsections with different settings, I think they can be summarized so that the readers can get a better overview. I suggest the authors to substantially polish this article. So I keep my score at current time.

---

> > > ### Author Response · Authors · 2022-08-25
> > > **Our focus is on benchmarking PPO and it's implementation in multi-agent settings rather than benchmarking MARL algorithms.**
> > >
> > > Thank you for the reply. To clarify the goal of our paper: we aim to benchmark the use of PPO in cooperative multi-agent settings, with a particular focus on **implementation details and hyperparameters relevant to PPO** ( similar to [1], which benchmarked PPO in single-agent RL ) -- our goal is not to study a variety of different algorithms in MARL.
> > >
> > > We would like to point out that the main algorithm part of the paper in Sec. 3 is short primarily because we make minimal algorithmic changes to PPO to adapt it to multi-agent settings - in fact, the primary difference is that the critic can receive global information in multi-agent settings (since the critic is not utilized at execution time). Everything else, including advantage estimation with GAE and policy/value-function update rules remain the same. We include a detailed algorithm box in the Appendix to fully demonstrate the implementation of our method.
> > >
> > > We find that it is implementation details and hyperparameter settings which are most crucial to getting PPO to perform strongly in cooperative multi-agent settings, and not algorithmic changes. The purpose of Sec. 5 is to empirically study what we find to be the most important factors in obtaining strong results with PPO in multi-agent settings.
> > >
> > > We compare PPO to different off-policy methods in each benchmark because each benchmark has its own established set of SOTA methods - for instance, RODE and QPLEX in SMAC, SAD in Hanabi, and TiKick and CDS in GRF. Hence, we compare to these methods in each of the corresponding benchmarks. We are currently running additional baseline methods in each setting (for instance, QPLEX and RODE in GRF and the MPEs), and will update our paper when these experiments finish.
> > >
> > > Overall, we aim to organize our paper in the following way: in Sec. 3, we give an overview of PPO and how it is algorithmically implemented in multi-agent settings (this is very similar to single-agent PPO). In Sec. 4, we present all of our primary results and show that PPO can achieve results comparable to the SOTA on a variety of benchmarks. In Sec. 5, we examine more specific implementation level factors and hyperparameter settings that influence PPO’s performance the most in multi-agent settings and give concrete suggestions as to what we find are best practices for each factor.
> > >
> > > [1] Andrychowicz, Marcin, Anton Raichuk, Piotr Stańczyk, Manu Orsini, Sertan Girgin, Raphaël Marinier, Leonard Hussenot et al. "What matters for on-policy deep actor-critic methods? a large-scale study." In International conference on learning representations. 2020.

---

> > > ### Author Response · Authors · 2022-08-27
> > > **We have reorganized the paper with changes in red.**
> > >
> > > We thank the reviewer for the valuable feedback and review. We have reorganized the manuscript according to the reviewer's suggestion on "MARL algorithms are compared in different experiment subsections with different settings". The detailed changes can be found in [the shared reply](https://openreview.net/forum?id=YVXaxB6L2Pl&noteId=hsqRTXoysoW). Please let us know if there is still ambiguity, and we will continue to improve the manuscript.

---

### Official Review · Reviewer_w7sL · 2022-07-25
**I do not think the workload, or the finding, or the idea is worth a scienfic research paper at NeurIPS.**

**Rating:** 4
**Confidence:** 5
**Correctness:** Yes.
**Clarity:** this paper is well written.

**Strengths:**

The authors provide the details of how to implement PPO on four benchmarks.

**Weaknesses:**

This work lacks scientific novelty. I do not think the performance of PPO in multiagent settings like SMAC is surprising. I have been using PPO in environments like SMAC long ago.
The followup work of this work has already been published elsewhere: https://arxiv.org/pdf/2206.07505.pdf


**Additional Feedback:**

A blog track might be the proper venue for this kind of work.

**Documentation:**

Yes.

**Ethics:**

No ethical concern.

**Relation To Prior Work:**

yes.

**Summary And Contributions:**

This paper implements PPO on four benchmarks and finds that PPO is a strong baseline for multiagent reinforcement learning.

---

> ### Author Response · Authors · 2022-08-13
> **We would like to remind the reviewer that we are under the benchmark track of NeurIPS**
>
> We strongly disagree with the reviewer’s complaint that our work lacks novelty. It is fully understandable that the reviewer uses PPO a lot — so do we. However, we respectfully would like to remind the reviewer that personal practices are not academic consensus. In fact, before the release of this benchmark, most SOTA results do not use PPO as a baseline [1, 2, 3]. This is exactly the reason we developed this benchmark work: we would like to make the community aware of the effectiveness of PPO.
>
> We also want to remind the reviewer that this is a benchmark paper whose purpose is to evaluate **existing** methods and present findings that people may not be aware of. We suggest the reviewer refer to last year’s published benchmark works. We just raise a few of them which are highly similar to our paper scope: MARL [4], Text-to-Image synthesis [5], Unsupervised RL [6], language [7], and time series [8].
>
> Moreover, according to the [call-for-paper page](https://neurips.cc/Conferences/2022/CallForDatasetsBenchmarks) of NeurIPS benchmark track, novelty isn’t the only criteria for evaluating the submissions. For example, **accessibility** is one aspect that is particularly mentioned. We would like to emphasize that it is because of our work’s accessibility that follow-up works (e.g. [9]) have been directly developed using our codebase and have been accepted in top venues. We believe that the fact that follow-up works built on this project fundamentally indicates that this work is significant and useful to the community.
>
> **References:**
>
> [1] Tabish Rashid, Mikayel Samvelyan, Christian Schroeder, Gregory Farquhar, Jakob Foerster, and Shimon Whiteson. QMIX: Monotonic value function factorisation for deep multi-agent reinforcement learning. In Jennifer Dy and Andreas Krause, editors, Proceedings of the 35th International Conference on Machine Learning, volume 80 of Proceedings of Machine Learning Research, pages 4295–4304. PMLR, 10–15 Jul 2018.
>
> [2] Tonghan Wang, Tarun Gupta, Anuj Mahajan, Bei Peng, Shimon Whiteson, and Chongjie Zhang. RODE: Learning roles to decompose multi-agent tasks. In International Conference on Learning Representations, 2021.
>
> [3] Jianhao Wang, Zhizhou Ren, Terry Liu, Yang Yu, and Chongjie Zhang. {QPLEX}: Duplex dueling multi-agent q-learning. In International Conference on Learning Representations, 2021.
>
> [4] Georgios Papoudakis, Filippos Christianos, Lukas Schäfer, and Stefano V Albrecht. Benchmarking multi-agent deep reinforcement learning algorithms in cooperative tasks. In Advances in Neural Information Processing Systems Track on Datasets and Benchmarks, 2021.
>
> [5] Dong Huk Park, Samaneh Azadi, Xihui Liu, Trevor Darrell, and Anna Rohrbach. Benchmark for Compositional Text to-Image Synthesis. In Thirty-Fifth Conference on Neural Information Processing Systems Datasets and Benchmarks Track (Round 1), 2021.
>
> [6] Laskin, Michael, Yarats, Denis, Liu, Hao, Lee, Kimin, Zhan, Albert, Lu, Kevin, Cang, Catherine, Pinto, Lerrel, and Abbeel, Pieter. URLB: Unsupervised reinforcement learning benchmark. In Thirty-fifth Conference on Neural Information Processing Systems Datasets and Benchmarks Track (Round 2), 2021.
>
> [7] Dumitrescu, S. D., Rebeja, P., Lorincz, B., Gaman, M., Avram, A., Ilie, M., Pruteanu, A., Stan, A., Rosia, L., Iacobescu, C., Morogan, L., Dima, G., Marchidan, G., Rebedea, T., Chitez, M., Yogatama, D., Ruder, S., Ionescu, R. T., Pascanu, R., and Patraucean, V. (2021). LiRo: Benchmark and leaderboard for Romanian language tasks. In Thirty-fifth Conference on Neural Information Processing Systems Datasets and Benchmarks Track (Round 1), 2021.
>
> [8] Kwei-Herng Lai, Daochen Zha, Junjie Xu, Yue Zhao, Guanchu Wang, and Xia Hu. 2021. Revisiting time series outlier detection: Definitions and benchmarks. In NeuraIPS (Datasets and Benchmarks Track).
>
> [9] Wei Fu, Chao Yu, Zelai Xu, Jiaqi Yang, Yi Wu. Revisiting Some Common Practices in Cooperative Multi-Agent Reinforcement Learning. In Proceedings of the 39th International Conference on Machine Learning, PMLR 162:6863-6877, 2022.

---

### Official Review · Reviewer_MGGd · 2022-07-27
**An exhaustive evaluation of PPO in cooperative MARL tasks and provides empirical suggestion.**

**Rating:** 7
**Confidence:** 4
**Correctness:** I believe the evaluation methods and …
**Clarity:** Yes.

**Strengths:**

The paper is easy to read and it contains multiple empirically useful suggestion to the PPO practitioner on how to setup their PPO training, each backed up with ablation study. The experiments contains fair comparisons, hyper parameter search and detailed ablation study on the tricks it suggests.

**Weaknesses:**

One of the limitation of the paper is the depth of analysis provided.

**Additional Feedback:**

How does the multi-headed actor-critic network structure affects the PPO’s performance?

**Documentation:**

Good. Source code is released.

**Relation To Prior Work:**

The PPO clipping can be performed in different ways in multi-agent cases, such as the [Coordinated Proximal Policy Optimization. zifan et al.].  Any benchmark experiments on clipping schemes?

**Summary And Contributions:**

This paper makes serval contributions:
1. Compare PPO’s  performance with off-policy baseline algorithm
2. Compare PPO’s sample efficiency with off-policy baseline algorithm
3. Provide tricks on how to setup training configs

---

> ### Author Response · Authors · 2022-08-13
> **Thank you for the review, here are our responses:**
>
> We thank the reviewer for the positive feedback and review. The primary goal of this work is to do a thorough empirical investigation into the performance and properties of PPO in cooperative multi-agent settings, and hence do not focus on theoretical analyses. We hope that our findings will motivate future work into investigating theoretical reasons behind the patterns we observe.
>
> - Regarding the clipping schemes, the paper [Coordinated Proximal Policy Optimization] is the follow-up work based on our paper and open-sourced code. We didn’t conduct experiments on clipping schemes in this paper and we will leave it as our future work.
>
> - Regarding the multi-headed actor-critic structure, we experimented with the impact of the architecture in SMAC but did not find any significant differences in performance on most maps. A possible reason is that network architecture is not a bottleneck for SMAC tasks and proper testbeds environments are needed to check the architecture influence, which we leave as our future work.

---

### Official Review · Reviewer_kxAg · 2022-07-28

**Rating:** 7
**Confidence:** 4
**Correctness:** Please see my comment about the evalu…

**Strengths:**

* The paper does a good job in evaluating PPO and comparing it against several baselines.
* The paper is in general well-written (besides some cases that I mention below).
* Section 5 does a good job in showing how different implementation details affect PPO's  performance.
* The code to reproduce the experiments is open-source.

**Weaknesses:**

My biggest concern in the paper is the evaluation consistency.
For example:
* line 104: "nearly all baseline methods utilize parameter sharing..." why not all of them?
* line 173: "The reported results for SAD and VDN are obtained from [15]." This makes the comparison inconsistent as different implementations lead to completely different achieved returns.
* Table 2: "Results are reported over at-least 3 seeds." Why not the same number of seeds?

Another concern is the suggestions of Section 5. In general, I liked these studies, BUT, the obtained conclusions are only applicable to the specific evaluation tasks. In general, I use PPO in my own MARL research, and I have seen many cases where the suggestions of the authors would result in lower performance. I would recommend to make it clear in the text that these suggestions will not always result in better performance, and that the findings are only applicable to the specific tasks.

* The definition of the Dec-POMDP in lines 91-97 is incorrect. The definition is missing the observation conditional probabilities function.

**Additional Feedback:**

* I recommend to the authors to fix the references as there are several inconsistencies.

* I recommend to the authors to increase the size of Figure 1, as it is difficult to read.
-----------------------------------------------------------

In general, the paper is well-known already in the community, counting more than 100 citations. It does a good job in evaluating PPO for MARL, and I believe that it should appear in the conference.

**Clarity:**

There are some language issues with phrasing. For example:
* line 10: "strong off-policy methods". what does strong mean in the context of the paper?
* line 11-12: rewards ---> returns (I have seen that several times in the paper)


**Documentation:**

Yes, code is on Github

**Relation To Prior Work:**

Yes

**Summary And Contributions:**

The authors evaluate multi-agent versions of PPO (IPPO and MAPPO) in four multi-agent environments, and compare it against several baselines. The authors perform several ablation studies to evaluate how different implementation details affect PPO's achieved returns.

---

> ### Author Response · Authors · 2022-08-13
> **Thank you for the review, here are our responses:**
>
> We thank the reviewer for the review and feedback. We have revised our paper to address several of the concerns you raise.
>
> - **Parameter sharing**: PPO and all baseline methods utilize parameter sharing in all situations with homogenous agents - the only case in which this is not true is in the MPE Speaker Listener scenario, in which the speaker and listener have different policies with separate parameters since the two agents have different observation and action spaces.
> - **SAD and VDN**: The training time and cost in Hanabi is significantly higher than the other benchmark environments. The code for SAD and VDN is developed by Facebook’s engineers, so we assume the code is well-tuned and optimized, which is confirmed by the authors of SAD and VDN. Our own re-implementations of SAD and VDN in our MAPPO codebase ran substantially slower with less sample efficiency than the numbers released in the original work, and we hence use the published numbers for a fair comparison. We are continuing our re-implementation efforts and will update our codebase accordingly when this is completed.
> - **Seeds**: The results of SAD and VDN use 10 seeds while MAPPO and IPPO use 3 seeds for games with more than 3 agents and 4 seeds for games with 2 agents due to the high training cost and limited compute time.
> - **Suggestions in Section 5**: While it is true that the suggestions we give in Section 5 are based on results from the 4 benchmark environments we evaluate in, we would like to point out that in our experiments, we apply all our suggestions and obtain strong performances over all the domains. Hence, we believe these suggestions can be a valid guide for tuning PPO in multi-agent settings. We are also running more ablation studies, which, however, takes a lot of time, and will post the results once they are ready. We will update our paper’s text to clarify this.

---

### Official Review · Reviewer_Xc16 · 2022-07-28
**Empirical Study for Shared Parameter PPO with Centralized-V in Cooperative Multi-Agent Games**

**Rating:** 5
**Confidence:** 4
**Clarity:** Yes.

**Strengths:**

1. This paper performs extensive experiments on the MPE, SMAC, Hanabi, and GRF environments compared with different MARL algorithms.
1. The authors ran ablations on their MAPPO implementation and gives tricky suggestions based on the results.

**Weaknesses:**

1. The results are not significant enough.
1. The paper is organized like a kind of blog or technical report rather than an academic paper. In this paper, the author simply lists the results and then gives the conclusion that "MAPPO outperforms ...". The experimental analysis is not sound enough.
1. Followed by the weakness above, the authors did not do any theoretical analysis of the results. The suggestions given in the paper may not effective in other settings where the paper was not tested.
1. In lines 33, 266, and 291, the authors use the term "we hypothesize that ...". The conclusions may need more evidence, e.g., theoretical analysis.
1. In this paper, the author only discussed MAPPO in cooperative environments with homogeneous agents. That is only a subset of cooperative multi-agent games. None of the heterogeneous settings are benchmarked.
1. Nearly all methods utilize parameter sharing, lack of results on non-sharing settings. Do the suggestions given by the author still hold without parameter sharing?
1. The authors discussed that PPO-based on-policy algorithms can achieve comparable performance to the off-policy methods with comparable sample efficiency. But for off-policy methods, the sampling and training stages can be executed in parallel. Are on-policy methods as time efficient as the off-policy methods?

**Additional Feedback:**

1. In this paper's title: "The Surprising Effectiveness of PPO in Cooperative Multi-Agent Games", but the author only studied the CTDE setting of MAPPO. The centralized methods are not discussed in this paper. It would be better to add the centralized PPO algorithm as an additional baseline.
2. The authors have tweaked the PDF page margin to fit the paper into 9 pages. Some tables and figures have gone too close to the page margin.

**Correctness:**

The experiment design is almost appropriate. But almost all experiments are tested in homogenous agent settings. None of the cooperative games with heterogeneous settings.

**Documentation:**

Yes. Provided with source code on GitHub.

**Relation To Prior Work:**

Yes.

**Summary And Contributions:**

This paper performs extensive experiments on MAPPO, i.e., shared parameter PPO with Centralized value function in cooperative multi-agent settings. The authors ran PPO-based MARL algorithms on the MPE, SMAC, Hanabi, and GRF environments compared to off-policy baselines. Demonstrate that PPO-based algorithms can achieve competitive performance to those state-of-the-art off-policy algorithms. Based on the results, the authors also give suggestions for implementation tricks for on-policy MARL algorithms.

---

> ### Author Response · Authors · 2022-08-13
> **We strongly disagree with the reviewer’s judgment**
>
> We appreciate the reviewer for acknowledging that “this paper performs extensive experiments”. We would first like to highlight to the reviewer that  there are a large number of works from last year's Benchmarks and Datasets track that follow the same paper organization as this one, **without theoretical analysis**. We simply list a few here [1, 2, 3, 4, 5]. We would like to emphasize that this is a benchmark paper, which is supposed to evaluate existing methods and present empirical findings. We leave theoretical analysis to other works, as has already been done in this year’s ICML [6].
>
> Regarding the scope of this paper, we have clearly stated multiple times that we focus on cooperative games, and we have tested our suggestions on four popular MARL testbeds in the existing literature. Our suggestions hold across all tested domains. Therefore, we disagree with the assessment that "The suggestions given in the paper may not be effective in other settings where the paper was not tested".
>
> We kindly remind the reviewer that, even beyond the benchmark track, there are many highly influential papers that do not have theoretical results — for example, the DQN paper [7]. Empirical evaluation is itself a widely accepted scientific practice and often guides important theory later on. We are also happy to test our presented techniques on more domains if the reviewer is willing to provide specific feedback.
>
> Finally, according to the [call-for-paper page](https://neurips.cc/Conferences/2022/CallForDatasetsBenchmarks) of the benchmark track, there are other aspects of a work, such as accessibility, which are important and impactful qualities. Our results have been widely reproduced and highly cited by many follow-up works such as [8, 9, 10, 11]. The GitHub repository is also well documented and under active maintenance. We hope the reviewer takes these factors into account in their review.
>
>
>
> **References:**
>
> [1] Georgios Papoudakis, Filippos Christianos, Lukas Schäfer, and Stefano V Albrecht. Benchmarking multi-agent deep reinforcement learning algorithms in cooperative tasks. In Advances in Neural Information Processing Systems Track on Datasets and Benchmarks, 2021.
>
> [2] Dong Huk Park, Samaneh Azadi, Xihui Liu, Trevor Darrell, and Anna Rohrbach. Benchmark for Compositional Text to-Image Synthesis. In Thirty-Fifth Conference on Neural Information Processing Systems Datasets and Benchmarks Track (Round 1), 2021.
>
> [3] Laskin, Michael, Yarats, Denis, Liu, Hao, Lee, Kimin, Zhan, Albert, Lu, Kevin, Cang, Catherine, Pinto, Lerrel, and Abbeel, Pieter. URLB: Unsupervised reinforcement learning benchmark. In Thirty-fifth Conference on Neural Information Processing Systems Datasets and Benchmarks Track (Round 2), 2021.
>
> [4] Dumitrescu, S. D., Rebeja, P., Lorincz, B., Gaman, M., Avram, A., Ilie, M., Pruteanu, A., Stan, A., Rosia, L., Iacobescu, C., Morogan, L., Dima, G., Marchidan, G., Rebedea, T., Chitez, M., Yogatama, D., Ruder, S., Ionescu, R. T., Pascanu, R., and Patraucean, V. (2021). LiRo: Benchmark and leaderboard for Romanian language tasks. In Thirty-fifth Conference on Neural Information Processing Systems Datasets and Benchmarks Track (Round 1), 2021.
>
> [5] Kwei-Herng Lai, Daochen Zha, Junjie Xu, Yue Zhao, Guanchu Wang, and Xia Hu. 2021. Revisiting time series outlier detection: Definitions and benchmarks. In NeuraIPS (Datasets and Benchmarks Track).
>
> [6] Wei Fu, Chao Yu, Zelai Xu, Jiaqi Yang, Yi Wu. Revisiting Some Common Practices in Cooperative Multi-Agent Reinforcement Learning. In Proceedings of the 39th International Conference on Machine Learning, PMLR 162:6863-6877, 2022.
>
> [7] Mnih, Volodymyr, Kavukcuoglu, Koray, Silver, David, Graves, Alex, Antonoglou, Ioannis, Wierstra, Daan, and Riedmiller, Martin. Playing atari with deep reinforcement learning. arXiv preprint arXiv:1312.5602, 2013.
>
> [8] Jakub Grudzien Kuba, Muning Wen, Yaodong Yang, Linghui Meng, Shangding Gu, Haifeng Zhang, David Henry Mguni, and Jun Wang. Settling the variance of multi-agent policy gradients. Advances in Neural Information Processing Systems, 2021.
>
> [9] Jianhong Wang, Wangkun Xu, Yunjie Gu, Wenbin Song, Tim C Green. Multi-Agent Reinforcement Learning for Active Voltage Control on Power Distribution Networks. Advances in Neural Information Processing Systems, 2021.
>
> [10] Dongsheng Ding, Chen-Yu Wei, Kaiqing Zhang, Mihailo Jovanovic. Independent Policy Gradient for Large-Scale Markov Potential Games: Sharper Rates, Function Approximation, and Game-Agnostic Convergence. Proceedings of the 39th International Conference on Machine Learning, PMLR 162:5166-5220, 2022.
>
> [11] Chi Jin, Qinghua Liu, Tiancheng Yu. The Power of Exploiter: Provable Multi-Agent RL in Large State Spaces. Proceedings of the 39th International Conference on Machine Learning, PMLR 162:10251-10279, 2022.

---

> > ### Comment · Reviewer_Xc16 · 2022-08-14
> > **The response does not resolve my concern. The suggestions given in the paper may not effective.**
> >
> > **The author's response does not answer my questions.** Does parameter-sharing affect the performance? What about the time cost? When will the suggestion hold and when will it fail?
> >
> > In line 91, the authors use the heterogeneous formulation of Dec-POMDP, i.e., each agent can have its own action space $\mathcal{A}_i$.
> > The authors claimed they would discuss heterogeneous agent settings in future work. But did not discuss the effect of parameter-sharing in homogenous settings, which may have a downside to the performance.
> >
> > > We would like to emphasize that this is a benchmark paper, which is supposed to evaluate existing methods and present empirical findings. We leave theoretical analysis to other works, as has already been done in this year’s ICML [6].
> >
> > In the reference ICML [6], the analysis shows that parameter sharing cannot learn an optimal policy for the XOR game, which is a simple multi-agent cooperative game with homogeneous agents. In this paper, almost all algorithms, including MAPPO and the baselines, are utilized with parameter sharing. **None of the non-shared settings are tested**, except MPE _comm_. I want to point out that, in the baseline algorithms, the original MADDPG does not use parameter sharing. But in the paper and the GitHub repo, **the authors modified the MADDPG with parameter sharing technique without any reason**.
> >
> > ------
> >
> > > Regarding the scope of this paper, we have clearly stated multiple times that we focus on cooperative games, and we have tested our suggestions on four popular MARL testbeds in the existing literature. Our suggestions hold across all tested domains. Therefore, we disagree with the assessment that "The suggestions given in the paper may not be effective in other settings where the paper was not tested".
> >
> > As the author posted:
> >
> > > Our suggestions hold across all tested domains.
> >
> > This comment would not resolve my question due to a lack of theoretical analysis. I'm asking for the "not tested" ones.
> >
> > >> The suggestions given in the paper may not be effective in other settings where the paper was not tested
> >
> > I think some of the suggestions given by the paper only hold in a narrow scope in MARL cooperative games.
> >
> > The global state should include all information about the environment state and the agent states. The agent can infer its agent-specific (AS) global state from the global state. In figure 3, the agent ID is included in the agent-specific (AS) global state but not in the environment-provided (EP) global state. Would this be a factor that AS is better than EP? If the policy parameters are not shared, would AS still be better than EP?
> >
> > In addition, in the reference "ICML [6]" the author commented, that shared policy cannot learn an optimal policy for the XOR game. XOR game is a fully cooperative game with homogenous agents, which is within the scope of the paper's focus. **The theoretical analysis gives us a suggestion: do not use parameter sharing.**
> >
> > [6] Wei Fu, Chao Yu, Zelai Xu, Jiaqi Yang, Yi Wu. Revisiting Some Common Practices in Cooperative Multi-Agent Reinforcement Learning. In Proceedings of the 39th International Conference on Machine Learning, PMLR 162:6863-6877, 2022.

---

> > > ### Author Response · Authors · 2022-08-14
> > > **Attempt to address reviewers concerns about scoping**
> > >
> > > We have updated paper (in red) to explicitly clarify that the experiment setting we focus on. More additional clarifications are as follows.
> > > We would like to clarify that parameter sharing does not mean that each agent's critic outputs the same value. As suggested in our paper, MAPPO uses agent-specific features, so different agent's critic values can be different in practice. In all our testbeds, the policy observation includes agent ID information as well. Note that an agent-ID conditioned neural network has the same representation power as separate ones. The referred ICML paper precisely states the equivalence and further shows that agent-ID conditioned policy/critic is essential for MAPPO's empirical success. Since the benchmark track does not require anonymity, we want to point out that **the referred ICML paper is from our group to serve as theoretical justification for why our MAPPO implementation works well in practice**. The claim that this paper indicates that parameter sharing should not be used is **not correct**; the paper referenced suggests the usage of shared parameters with agent-IDs as part of the features so we are unclear why this paper is used as a criticism of our work
> > >
> > > We understand that there can be limitations to parameter sharing (as we point out in our ICML paper that is quoted here) but its use here should not be viewed as a limitation of the paper but is purposefully used to ensure a fair comparison with recent literature. Parameter-sharing is a standard implementation in most recent MARL works, including the hide-and-seek project, SAD, QMIX, QPLEX, and RODE (please refer to line 104). We follow the same implementation for a fair comparison, and that's why we change the original MADDPG implementation to parameter-sharing as well. As what we have stated in the paper, we have made sure our re-implementations are at least as good as the original papers. The MADDPG codebase has been there for 5 years. It is the responsibility of a good benchmark paper to tune all the baselines well using modern tricks.
> > >
> > > The reviewer seems to be concerned that we are claiming that because MAPPO works well in these domains, it will work in other domains. This is NOT a claim made in the text of this paper and we are happy to correct the writing if the reviewer can point to areas where this impression may have come across; **the only claims made in this paper are that on the entire range of standard cooperative MARL benchmarks, MAPPO is performing well and that certain hyperparameter settings are essential to achieving this performance**. The reviewer is correct to note that these benchmarks share features such as having the concatenated agent state contain most of the relevant information. It would be interesting to generate new, standard benchmarks that do not have these features but this seems outside the scope of this paper.
> > >
> > > Regarding the scope, it is true for __any empirical work__ that "*The suggestions given in the paper may not be effective in other settings where the paper was not tested.*" It is always possible to ask more questions as to how the results would generalize to new domains, but we want to emphasize that this is exactly the merit of good empirical work: it brings novel insights and raises new questions for the community to guide later research and push the frontier of human knowledge. Our work touches on a specific subset of possible parameter-sharing settings and MARL games and is intended to establish rigorous work within that domain. We are happy that the reviewer has new questions based on our submission and we believe these thoughts could lead to insightful follow-up works, as what we have done in this year's ICML. Nevertheless, we would kindly request that the reviewer judge our submission based on its current scope.

---

> > > > ### Comment · Reviewer_w7sL · 2022-08-14
> > > > **Can we draw the conclusion from your ICML paper that PPO is effective in multiagent settings, at least some of them?**
> > > >
> > > > Can we draw the conclusion from your ICML paper that PPO is effective in multiagent settings, at least some of them?  I believe the answer is yes. Then why is this current paper still surprising? If not, why is it publishable?

---

> > > > > ### Author Response · Authors · 2022-08-14
> > > > > **This paper is empirical backing used extensively in other works**
> > > > >
> > > > > This paper provides the empirical backing that the ICML paper stands upon and the ICML paper is at least partially an investigation of the parameter sharing performed in this work. The ICML paper makes explicit that it is a validation for the empirical findings in this and related works. Without the context of this paper, the ICML paper would not make sense. That this paper and its follow-up paper would be published out of order does not seem like a mark to hold against this paper under the reviewing guidelines.

---

> > ### Comment · Reviewer_Xc16 · 2022-08-14
> > **More experiments and analysis should be added**
> >
> > > We appreciate the reviewer for acknowledging that “this paper performs extensive experiments”. We would first like to highlight to the reviewer that there are a large number of works from last year's Benchmarks and Datasets track that follow the same paper organization as this one, without theoretical analysis. We simply list a few here [1, 2, 3, 4, 5]. We would like to emphasize that this is a benchmark paper, which is supposed to evaluate existing methods and present empirical findings. We leave theoretical analysis to other works, as has already been done in this year’s ICML [6].
> >
> > If the authors claim the paper is a benchmark paper, they should demonstrate (like [1] did):
> > - Which baselines and testbeds are chosen and why? Do they cover most cases?
> > - What are the differences between the baselines and the testbeds?
> > - How the evaluations are designed?
> > - How the codebase is organized?
> > - The results and findings.
> >
> > If the authors claim the main contribution of the paper is the implementation details of MAPPO, I still think the paper needs to add more experiments to the details of the implementation. *Especially, the new findings when extending the single-agent PPO to multi-agent PPO.* For example, parameter-sharing is also part of the implementation of MAPPO. The authors need to discuss it in the paper. That would need more experiments, such as the non-shared settings, to support the authors' findings. As the MAPPO is the natural extension of PPO in multi-agent settings, some of the suggestions for the single-agent PPO are already made in [7]. They should not be considered as the contribution of this paper, such as _Value Normalization (VN)_. Unless the authors can tell the multi-agent VN is different from the single-agent VN (suggested in [7]).
> >
> > ------
> >
> > Reference:
> >
> > [1] Georgios Papoudakis, Filippos Christianos, Lukas Schäfer, and Stefano V Albrecht. Benchmarking multi-agent deep reinforcement learning algorithms in cooperative tasks. In Advances in Neural Information Processing Systems Track on Datasets and Benchmarks, 2021.
> >
> > [2] Dong Huk Park, Samaneh Azadi, Xihui Liu, Trevor Darrell, and Anna Rohrbach. Benchmark for Compositional Text to-Image Synthesis. In Thirty-Fifth Conference on Neural Information Processing Systems Datasets and Benchmarks Track (Round 1), 2021.
> >
> > [3] Laskin, Michael, Yarats, Denis, Liu, Hao, Lee, Kimin, Zhan, Albert, Lu, Kevin, Cang, Catherine, Pinto, Lerrel, and Abbeel, Pieter. URLB: Unsupervised reinforcement learning benchmark. In Thirty-fifth Conference on Neural Information Processing Systems Datasets and Benchmarks Track (Round 2), 2021.
> >
> > [4] Dumitrescu, S. D., Rebeja, P., Lorincz, B., Gaman, M., Avram, A., Ilie, M., Pruteanu, A., Stan, A., Rosia, L., Iacobescu, C., Morogan, L., Dima, G., Marchidan, G., Rebedea, T., Chitez, M., Yogatama, D., Ruder, S., Ionescu, R. T., Pascanu, R., and Patraucean, V. (2021). LiRo: Benchmark and leaderboard for Romanian language tasks. In Thirty-fifth Conference on Neural Information Processing Systems Datasets and Benchmarks Track (Round 1), 2021.
> >
> > [5] Kwei-Herng Lai, Daochen Zha, Junjie Xu, Yue Zhao, Guanchu Wang, and Xia Hu. 2021. Revisiting time series outlier detection: Definitions and benchmarks. In NeuraIPS (Datasets and Benchmarks Track).
> >
> > [6] Wei Fu, Chao Yu, Zelai Xu, Jiaqi Yang, Yi Wu. Revisiting Some Common Practices in Cooperative Multi-Agent Reinforcement Learning. In Proceedings of the 39th International Conference on Machine Learning, PMLR 162:6863-6877, 2022.
> >
> > [7] Marcin Andrychowicz, Anton Raichuk, Piotr Stańczyk, Manu Orsini, Sertan Girgin, Raphaël Marinier, Leonard Hussenot, Matthieu Geist, Olivier Pietquin, Marcin Michalski, Sylvain Gelly, Olivier Bachem. What Matters for On-Policy Deep Actor-Critic Methods? A Large-Scale Study. In Proceedings of the International Conference on Learning Representations 2021 (Oral).

---

> > > ### Author Response · Authors · 2022-08-19
> > > **We have updated the paper and appendix to address reviewer‘s issue**
> > >
> > > Thanks for the review and feedback. We have updated paper and appendix to address reviewer's issue. Here are our responses:
> > > - Regarding your first set of points:
> > >
> > >   1. Which baselines and testbeds are chosen and why? Do they cover most cases?
> > >
> > >   We choose the testbeds because they cover a diverse set of commonly studied benchmarks in cooperative settings. The baselines are correspondingly chosen to be the SOTA in each testbed (e.g. RODE in SMAC, CDS in GRF) - we also evaluate other popular methods in each testbed (e.g. QMix / QPlex in SMAC).
> > >
> > >   2. What are the differences between the baselines and the testbeds?
> > >
> > >   Although all testbeds study cooperative settings, they differ in game dynamics. For instance, the MPEs have a set of navigation and communication tasks, SMAC requires agents to simultaneously coordinate to defeat a set of enemies and requires strategies such as teaming up on one enemy at a time, Hanabi has unique turn-based dynamics which requires agents to convey information through actions, and GRF requires agents to coordinate and take on different roles in the game of football.
> > >
> > >   3. How the evaluations are designed?
> > >
> > >   The evaluations are organized as described in Sections 4.1 - 4.4. In each testbed, we obtain evaluations results as has been in the past works for each testbed - for instance, for the MPEs, we report the reward curves over training, averaged over 10 seeds. In SMAC, throughout the training process, we evaluate over 32 games and compute the success rate by averaging over those 32 games.To compute the final success rate for the seed, we take the success rates over the final 10 evaluations during training and compute the median of those values. The final reported value is the median of each seed’s success rate. In Hanabi, we follow [1, 2] and report the average returns across at-least 3 random seeds as well as the best score achieved by any seed. The returns are averaged over 10k games. In GRF, we follow CDS [3]; we compute the success rate over 100 rollouts of the game and report the mean success rate over the last 10 evaluations, averaged over random seeds.
> > >
> > >   4. How the codebase is organized?
> > >
> > >   Because of space constraints in our paper, we do not describe the codebase’s organization. However, the detailed READMEs in our GitHub repo (https://github.com/marlbenchmark) contains information about how to run our code and its structure.
> > >
> > >   5. The results and findings.
> > >
> > >   The results and findings are summarized in detail in Sections 4 and 5, as well as the Appendix.
> > >
> > > - We utilize parameter sharing when there are homogenous agents because this has been the accepted practice in the majority of past MARL works - for instance [4, 5, 6, 7] all utilize parameter sharing as this simplifies the learning process, since there are fewer overall parameters to learn We have clarified this in the intro, and results sections of our paper. We focus on other aspects of MAPPO (critic input representation, mini-batch size, number of epochs, etc) since these aspects are understudied in MARL settings while past works have analyzed the impact of parameter sharing [8,9]. Specifically, we revisit this practice in [9] (Tables 4 and 5) - we have updated our appendix C.2 to include these results for easy access.
> > >
> > > - Regarding the contributions of our suggestions - it is true that [7] studied value normalization in single agent RL and found that it at times aids performance. We include our suggestion about value-normalization because it has a significant positive impact on performance in several settings (e.g. the MPEs) and never hurts. This differs from the findings of [7], which state that value normalization sometimes helps and sometimes hurts performance. Our other suggestions are specific to using PPO in multi-agent settings - for instance, our suggestions regarding clipping, training data usage, and batch-size differ from common practice in single-agent RL, which we believe are particularly valuable suggestions to the community.
> > >
> > > - Finally, we would like to emphasize that our paper has several contributions in addition to our suggestions regarding the implementation details of MAPPO - specifically, our demonstration that MAPPO achieves scores competitive to SOTA off-policy methods in a variety of cooperative environments, and an actively maintained GitHub repo which implements MAPPO and several other off-policy methods.

---

> > > > ### Author Response · Authors · 2022-08-19
> > > > **References**
> > > >
> > > > **References**:
> > > >
> > > > [1] Jakob Foerster, Francis Song, Edward Hughes, Neil Burch, Iain Dunning, Shimon Whiteson, Matthew Botvinick, and Michael Bowling. Bayesian action decoder for deep multi-agent reinforcement learning. In International Conference on Machine Learning, pp. 1942–1951, 2019.
> > > >
> > > > [2] Hengyuan Hu and Jakob N Foerster. Simplified action decoder for deep multi-agent reinforcement learning. In International Conference on Learning Representations, 2020.
> > > >
> > > > [3] Chenghao Li, Tonghan Wang, Chengjie Wu, Qianchuan Zhao, Jun Yang, and Chongjie Zhang. Celebrating diversity in shared multi-agent reinforcement learning, 2021.
> > > >
> > > > [4] Filippos Christianos, Georgios Papoudakis, Arrasy Rahman, and Stefano V. Albrecht. Scaling multi-agent reinforcement learning with selective parameter sharing, 2021.
> > > >
> > > > [5] Tabish Rashid, Mikayel Samvelyan, Christian Schroeder, Gregory Farquhar, Jakob Foerster, and Shimon Whiteson. QMIX: Monotonic value function factorisation for deep multi-agent reinforcement learning. In Jennifer Dy and Andreas Krause, editors, Proceedings of the 35th International Conference on Machine Learning, volume 80 of Proceedings of Machine Learning Research, pages 4295–4304. PMLR, 10–15 Jul 2018.
> > > >
> > > > [6] Tonghan Wang, Tarun Gupta, Anuj Mahajan, Bei Peng, Shimon Whiteson, and Chongjie Zhang. RODE: Learning roles to decompose multi-agent tasks. In International Conference on Learning Representations, 2021.
> > > >
> > > > [7] Jianhao Wang, Zhizhou Ren, Terry Liu, Yang Yu, and Chongjie Zhang. {QPLEX}: Duplex dueling multi-agent q-learning. In International Conference on Learning Representations, 2021.
> > > >
> > > > [8] J. K. Terry, Nathaniel Grammel, Ananth Hari, Luis Santos, and Benjamin Black. Revisiting parameter sharing in multi-agent deep reinforcement learning, 2021.
> > > >
> > > > [9] Wei Fu, Chao Yu, Zelai Xu, Jiaqi Yang, Yi Wu. Revisiting Some Common Practices in Cooperative Multi-Agent Reinforcement Learning. In Proceedings of the 39th International Conference on Machine Learning, PMLR 162:6863-6877, 2022.

---

> ### Comment · Reviewer_Xc16 · 2022-08-20
> **Thanks for the clarification**
>
> The authors add a section for parameter-sharing the supplementary material. Regarding the source code repo, it is really helpful to the multi-agent learning community. I have increased my rating to 5. It's a great work but I still think the paper itself is not well organized for a benchmark track. So I keep my negative score.

---

> > ### Author Response · Authors · 2022-08-27
> > **We have reorganized the paper with changes in red**
> >
> > We appreciate the reviewer for acknowledging that "It's a great work" and we have reorganized the manuscript according to the reviewer's suggestion. The detailed changes can be found in [the shared reply](https://openreview.net/forum?id=YVXaxB6L2Pl&noteId=hsqRTXoysoW). Please let us know if there is still ambiguity, and we will continue to improve the manuscript.

---

### Author Response · Authors · 2022-08-13
**Paper updated with changes in red**

We have updated our paper and appendix to incorporate all the feedback from reviewers. All the changes are colored red.

---

### Author Response · Authors · 2022-08-27
**We have reorganized the paper according to the reviewers’ suggestions.**

We thank the suggestions from reviewers F2xg, Xc16, Ryeg that our paper needs further re-organization for a better view. We have reorganized the manuscript with changes in red. The main changes include:

- Reorganize Sec. 3 “PPO in multi-agent settings". We separate this section into three parts: 3.1 Preliminary, 3.2 description of MAPPO & IPPO as well as their differences, and 3.3 Implementation details. We further describe the parameter-sharing setting in Sec. 3.3.
- Reorganize Sec. 4 ”Main Results”.
  - We add a separate part “4.1 Testbeds and Baselines” to describe the chosen testbeds and all adopted baselines. We further describe the **common** experimental settings we adopted for all off-policy baselines, including the hyper-parameter search, parameter-sharing setting, and training compute.
  - We re-structure Sec. 4.2-4.5 into two parts, the experiment setting and experiment results. The experiment setting part mainly describes the setting **specific** to each testbed and the evaluation method. The experiment results part analyzes the results.

---

### Meta-Review · Area_Chair_8Hz6 · 2022-09-06

**Recommendation:** Accept
**Confidence:** 4

**Metareview:**

The paper presents extensive evidence showing PPO to be a (surprisingly) strong algorithm in multi-agent RL environments compared to more sophisticated and specialized approaches. The paper provides a helpful codebase along with elaboration on a variety of implementation details which significantly matter for PPO's good performance.

The reviewers had conflicting views of the paper. I'm glad to see that the submission authors used the review period wisely to significantly improve the clarity and organization of the paper, which was one of the major concerns (thanks to the reviewers for raising these concerns!). Even so, while most reviewers appreciated the extensive and detailed experiments and ablations of the paper, concerns lingered regarding novelty and significance of the work. In response to these concerns, I have consulted with a few domain experts in the field of multi-agent RL. Through these consultations, I have determined that the submission is indeed impactful, with its findings already being extensively used and built upon. Thus, altogether I believe this submission meets the bar for acceptance.

---

### Decision · Program_Chairs · 2022-09-16

Accept